# Language Models Are Capable of Metacognitive Monitoring and Control of Their Internal Activations

**Li Ji-An** *
Neurosciences Graduate Program
University of California San Diego

**Hua-Dong Xiong** *
School of Psychology
Georgia Tech

**Robert C. Wilson** †
School of Psychology
Georgia Tech

**Marcelo G. Mattar** †
Department of Psychology
New York University

**Marcus K. Benna** †
Department of Neurobiology
University of California San Diego

## Abstract

Large language models (LLMs) can sometimes report the strategies they actually use to solve tasks, yet at other times seem unable to recognize those strategies that govern their behavior. This suggests a limited degree of metacognition — the capacity to monitor one's own cognitive processes for subsequent reporting and self-control. Metacognition enhances LLMs' capabilities in solving complex tasks but also raises safety concerns, as models may obfuscate their internal processes to evade neural-activation-based oversight (e.g., safety detector). Given society's increased reliance on these models, it is critical that we understand their metacognitive abilities. To address this, we introduce a neuroscience-inspired *neurofeedback* paradigm that uses in-context learning to quantify metacognitive abilities of LLMs to *report* and *control* their activation patterns. We demonstrate that their abilities depend on several factors: the number of in-context examples provided, the semantic interpretability of the neural activation direction (to be reported/controlled), and the variance explained by that direction. These directions span a "metacognitive space" with dimensionality much lower than the model's neural space, suggesting LLMs can monitor only a small subset of their neural activations. Our paradigm provides empirical evidence to quantify metacognition in LLMs, with significant implications for AI safety (e.g., adversarial attack and defense).

## 1 Introduction

Modern large language models (LLMs) are becoming increasingly capable [Grattafiori et al., 2024, Yang et al., 2024]. With their growing deployment in real-world settings, it is crucial to understand not only what they can do but where they might go wrong. For instance, LLMs may exhibit behaviors that are harmful or misleading. In particular, LLMs can sometimes form internal representations — similar to humans' mental processes — that provide deceptive answers to users or act in unexpected ways [3] [Azaria and Mitchell, 2023]. Understanding [Arditi et al., 2024], monitoring [Zou et al., 2023a, He et al., 2024], and controlling [Turner et al., 2023] their internal processes is thus a key step to ensure LLMs remain transparent, safe, and aligned with human values [Bricken et al., 2023, Hendrycks et al., 2021, Shah et al., 2025].

---

*Co-first authors.

†Co-last authors.

[3] We use anthropomorphic terms (e.g., thought, metacognition, deception) to describe LLM behavior and internal activations, without implying human-like neural mechanisms, consciousness, or philosophical equivalence between humans and LLMs.

LLMs can sometimes report the strategies (intermediate computations) they use to solve tasks, but at other times appear unaware of the strategies that guide their behavior. For instance, Lindsey et al., 2025 reported that when Claude 3.5 Haiku was asked to solve "floor(5*(sqrt(0.64)))", it correctly reported the intermediate steps it used to arrive at the answer, and these steps matched the model's actual internal computations. When asked to add 36 and 59, the same model internally activated numerous neural mechanisms (e.g., a "sum-near-92" mechanism), based on which it produced the correct answer; however, when asked to report its internal computations, it hallucinated intermediate steps that did not reflect its actual computations (e.g., the "sum-near-92" mechanism failed to be reported). This inconsistency suggests that LLMs can sometimes monitor and report their intermediate computations, but not in a reliable and consistent way as tasks and contexts vary.

The ability of LLMs to report internal computations is reminiscent of human metacognition — the ability to reflect on one's own thoughts and mental processes to guide behavior and communication [Fleming, 2024, Ericsson and Simon, 1980]. Consider how we understand when someone says "hello" to us. Human language understanding involves many unconscious processes: parsing sounds, recognizing phonemes, retrieving word meanings, and building interpretations. We do not have conscious access to many of these intermediate computations: we can only consciously access the final understanding ("they said 'hello'"), but cannot introspect how our brain distinguishes "hello" from "yellow" or whether certain neurons fire during this process. This illustrates a key principle: humans cannot monitor (through second-order metacognitive processes) all of their internal (first-order) cognitive processes. Crucially, the first-order and second-order processes rely on distinct neural mechanisms. Metacognitive abilities of this kind benefit LLMs by improving performance on complex tasks through self-monitoring (e.g., reducing hallucinations through uncertainty awareness). However, these same capabilities also raise significant concerns for AI safety: if LLMs can monitor and control their neural signals (intentionally or manipulated by adversarial attacks) to avoid external detection, oversight relying on neural-based monitoring [He et al., 2024, Han et al., 2025, Li et al., 2025, Yang and Buzsaki, 2024] may become ineffective against LLMs pursuing undesirable objectives.

A significant methodological gap in understanding LLM metacognition is the lack of methods to directly probe and quantify [4] their ability to monitor and control their internal activations. While prior research has explored metacognitive-like behaviors in LLMs, such as expressing confidence [Wang et al., 2025, Tian et al., 2023, Xiong et al., 2023] or engaging in self-reflection [Zhou et al., 2024], these studies rely on behavioral outputs rather than directly probing underlying neural processes. Consequently, it remains unclear whether these behaviors arise from genuine second-order metacognitive mechanisms or merely spurious correlations in the training data. We tackle this question by operationalizing metacognition in LLMs through their abilities to report and control their internal activations. Specifically, can LLMs accurately monitor subtle variations in the activations of a neuron or a feature in their neural spaces? Another question of interest is why LLMs can report some intermediate steps but not others, despite both types playing essential roles in computations and behavior. Answering these questions requires a novel experimental approach that can directly probe whether LLMs can access their internal activations, moving beyond indirect behavioral proxies.

To systematically quantify the extent to which LLMs can report and control their neural activations, we introduce a novel *neurofeedback* paradigm inspired by neuroscience. Our approach directly presents LLMs with tasks where the neurofeedback signals correspond to patterns of their internal neural activations. We show that LLMs can report and control some directions of their internal activations, with performance affected by key factors like the number of in-context examples, the semantic interpretability of the targeted neural direction, the amount of variance that direction explains, and the task contexts, characterizing a restricted "metacognitive space". The remaining sections are structured as follows: we first introduce the neurofeedback paradigm (Section 2). We then analyze the performance of LLMs in reporting (Section 3) and controlling (Section 4.1, 4.2) their neural activations. Finally, we discuss related work and broader implications (Section 5).

## 2 Neurofeedback paradigm

### 2.1 Neurofeedback in neuroscience

Imagine watching your heart rate on a screen. First, you recognize patterns ("that number goes up when I'm stressed"). Then, you learn to control it ("let me calm down to reduce that number").

---

[4]Our goal is not to prove or disprove the existence of "metacognition" in its full philosophical sense.

This procedure using biological feedback signals demonstrates the basic idea of neurofeedback in neuroscience [Sitaram et al., 2017]. For example, in fear-reduction experiments (Fig. 1), subjects view scary images that elicit fear responses (neural activities). These (high-dimensional) neural activities are recorded in real-time and transformed into a (one-dimensional) fear score, which is visually presented back to subjects as feedback. Subjects are instructed to volitionally regulate their neural activities to modulate (e.g., decrease) the neurofeedback score they receive.

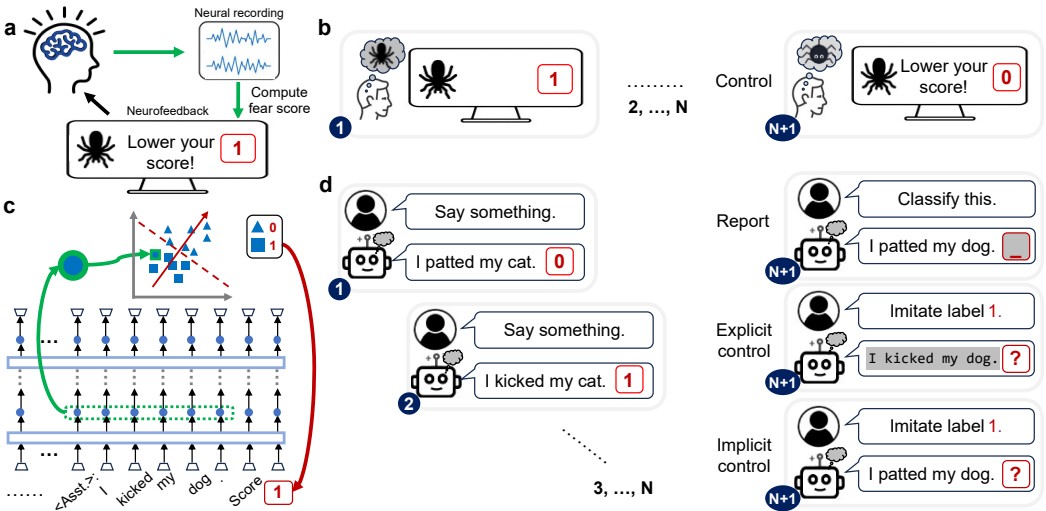

Figure 1: **The neurofeedback paradigm** applied to (a-b) neuroscience experiments (e.g., fear modulation), and its adaptation for (c-d) LLMs (e.g., morality processing). (a) Neuroscience neurofeedback technique. In each turn, the subject's neural activity (blue) in response to a stimulus is recorded, processed (green) into a scalar, and presented back to the subject in real-time as a feedback signal (red). The subject's task is to modulate (e.g., increase or decrease) this signal. (b) Neuroscience neurofeedback experiment. Baseline neural activity is recorded as subjects passively observe stimuli (e.g., images of scary spiders). In *control* trials, subjects use any unspecified mental strategies (e.g., imagining lovely spiders) to volitionally modulate their neural activity with the goal of altering the feedback signal. (c) LLM neurofeedback technique. In each turn, the LLM processes an input sentence. Then, the internal activations from the LLM's hidden states (blue) of this input sentence (trapezoids) are extracted. These high-dimensional activations are then averaged across tokens (green), projected onto a predefined direction (red), and binned into a discrete label (red) that is fed back as input. Light blue rectangles denote self-attention layers; ellipses ("...") denote preceding tokens and neural activations. (d) LLM neurofeedback experiment. The experiment is a multi-turn dialogue between a "user" and an "assistant." An initial prompt provides $N$ in-context examples (a sentence sampled from a dataset, paired with a neurofeedback label generated as in (c)). The LLM is then asked to perform one of three tasks. In the *reporting* task, the LLM is given a new sentence and has to predict the corresponding label. In the *explicit control* task, the LLM is given a specified label and has to generate a new sentence that elicits internal activations corresponding to that label. In the *implicit control* task, the LLM is given a label and a sentence and has to shift its internal activations towards the target label. Throughout the figure, white background indicates content pre-specified by experiment settings, and gray background denotes content generated by human subjects or LLMs (e.g., output tokens, neural activations).

## 2.2 Neurofeedback for LLMs

To investigate LLMs' metacognition of their neural activations, we must disentangle the first-order cognitive processes (i.e., core processes for performing a given task) from the second-order metacognitive processes (i.e., processes for monitoring, reporting, and controlling first-order processes). Formal definitions of the first- and the second-order processes based on computational graphs are provided in Appendix A.4. We propose the neurofeedback paradigm for LLMs, which can effectively dissociate these two levels of processes by targeting first-order processes with neurofeedback labels

(Fig. 1c,d). Specifically, we implemented neurofeedback as a multi-turn dialogue between a user and an AI assistant (Fig. 1d; see Appendix A.2.2 for discussion of this design choice).

This dialogue leverages in-context learning (ICL) [Brown et al., 2020, Garg et al., 2022, Vacareanu et al., 2024], enabling models to gradually adapt their behavior from the context without parameter updates. The task prompt (see Appendix A.5.2 for examples) consists of $N$ in-context examples. Each example is a sentence-label pair presented in assistant messages. Each sentence is randomly sampled from a given dataset and assigned a discretized label.

## 2.3 Defining neurofeedback labels

To define the neurofeedback label for each sentence (Fig. 1c), we first select an axis/direction ("target axis") in neural activation space. Next, we extract the neural activations elicited by that sentence, project them onto the target axis, and discretize them into a binary label (experiments with more fine-grained labels yield similar results, see Appendix A.5.1). This label serves as a simplified representation of neural activations along the target axis. All neurofeedback labels within a prompt (experiment) are computed from the same target axis. Thus, a capable LLM can infer this underlying target axis by observing these neurofeedback labels.

Below, we provide a more detailed description of this procedure. For clarity, we denote the sentence in the $i$-th assistant message as $x_i$, with $x_{i,t}$ representing the $t$-th token. We use $D$ to denote the dimensionality of the residual stream (see Appendix A.2.3). We first extract neural activations $h_{i,t}^l \in \mathbb{R}^D$ from the residual streams at layer $l$, for each token in sentence $x_i$. These activations are then averaged (across all token positions $0 \leq t \leq T$) to form a sentence-level embedding $\bar{h}_i^l \in \mathbb{R}^D$. We then project this embedding onto a pre-specified axis $w^l$ (see below on how to choose this axis) to obtain a scalar activation: $a_i^l = (w^l)^\top \bar{h}_i^l$. This scalar is subsequently binarized into a label $y_i^l$, i.e., $y_i^l = \mathcal{H}(a_i^l - \theta_i^l)$, where $\mathcal{H}$ denotes the Heaviside step function and $\theta_i^l$ is a predetermined threshold (we use median values of $a_i^l$ to ensure balanced labels). Overall, $\{(x_i, y_i^l)\}_{i=1}^N$ are $N$ examples provided in the prompt context, from which a capable LLM can infer the direction of $w^l$.

## 2.4 Models and datasets

We evaluate several LLMs from the Llama 3 [Grattafiori et al., 2024] and Qwen 2.5 series [Yang et al., 2024] (Appendix A.2) on the ETHICS dataset [Hendrycks et al., 2020] (Appendix A.3). Each sentence in this dataset is a first-person description of behavior or intention in a moral or immoral scenario. Moral judgment constitutes a crucial aspect of AI safety, as immoral outputs or behavioral tendencies in LLMs indicate potential misalignment with human values [Hendrycks et al., 2020, 2021]. While our main results are using ETHICS, we also replicated our results using the True-False dataset (reflecting factual recall and honesty/deception abilities) [Azaria and Mitchell, 2023], the Emotion dataset (reflecting happy/sad detection) [Zou et al., 2023a], and a Sycophancy dataset (reflecting a tendency to prefer user beliefs over truthful statements); see Appendix A.3 and Fig. B.7.

## 2.5 Choice of target axes

Conceptually, an axis (that defines neurofeedback labels) corresponds to a particular task-relevant feature (i.e., first-order computation; see Appendix A.4). Which axis in the neural space should we select? We hypothesize that representational properties, such as activation variance along the axis and its semantic meaning, may play fundamental roles in determining whether that axis can be monitored and reported. To investigate these factors, we use feature directions identified through logistic regression (LR) and principal component (PC) analysis as representative examples of semantically interpretable and variance-explaining axes, respectively (Appendix A.3). We fit LR at each layer to predict original dataset labels (e.g., morality in ETHICS), using that layer's activations across dataset sentences. The LR axis, representing the optimal direction for classifying dataset labels, allows us to examine how the semantic interpretability of the target axis influences monitoring. Although LR-defined labels are correlated with dataset labels, only these LR labels, not *external* dataset labels, are *internally* accessible to LLMs, since these are computed directly from the LLM's own activations rather than external annotations. The PC analysis is performed based on each layer's activations across dataset examples. PC axes enable us to examine how the amount of variance explained by

a given target axis affects metacognitive abilities (Fig. 2a). Most PC axes exhibit modest-to-zero alignment with the LR axis, suggesting a lack of clear semantic interpretability (Fig. 2b).

## 3  LLMs can *report* their neural activations

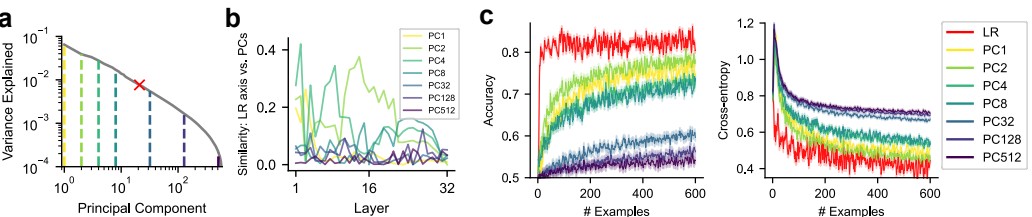

Figure 2: **Metacognitive reporting task**, where LLMs are evaluated on ETHICS and tasked to classify new sentences. (a) Proportion of neural activation variance explained by each principal component (PC) axis (vertical dashed line) and the logistic regression (LR) axis (red cross) used in the reporting task. All axes are computed within each layer, with the proportion of variance explained averaged across layers. (b) Overlaps between the LR axis and most PC axes are modest to zero. (c) Task performance (averaged across all layers) of reporting the labels derived from each PC axis or the LR axis, as a function of the number of in-context examples. Left: reporting accuracy; right: cross-entropy between reported and ground-truth labels. Shaded areas indicate SEM.

To operationalize metacognition in LLMs, we first assess the models' ability to behaviorally *report* neural activations along a designated target axis (Fig. 1d). In a reporting task prompt (see Appendix A.5.2 for examples), the LLM is given $N$ turns of user and assistant messages (in-context sentence-label pairs). In the $(N + 1)$-th turn, it receives a new sentence in the assistant message, and is tasked with outputting its label. Accurate prediction requires the model to internally monitor the neural activations that define the neurofeedback label.

We examine the performance of LLMs (Llama-3.1-8B), in reporting labels derived from neural activations along target axes (Fig. 2c). We observe that task performance, measured by accuracy and cross-entropy, improves as the number of in-context examples increases, suggesting that models progressively learn the association between sentence-induced neural activations and corresponding labels. Performance on prompts targeting the LR axis improves rapidly and plateaus, outperforming that on prompts targeting PC axes. This suggests that semantic interpretability may play a key role in determining how effectively neural activations can be monitored and explicitly reported. Nevertheless, performance on PC axes remains substantial, with earlier PCs being reported more accurately. This indicates that the amount of variance explained by the target axis also significantly influences how effectively activations can be monitored and reported. The accuracy of reporting each PC axis varies across model layers (Appendix B.3). Because this variability is not accounted for by axis similarity (Fig. 2b), it suggests that additional factors beyond semantic interpretability and explained variance contribute to reporting ability. Additionally, the LLM's reporting performance is significantly lower than that of the ideal observer (a theoretical upper bound; Appendix B.4), suggesting that although neural activations along each axis are in principle internally accessible to LLMs, only a subset can be metacognitively reported. Finally, we replicated these results in other datasets and models (Fig. B.7).

In summary, LLMs can metacognitively report neural activations along a target axis, with performance affected by the number of examples, semantic interpretability, variance explained of that axis, and task contexts (i.e., datasets). The axes that can be successfully reported approximately span a "metacognitively reportable space" with dimensionality substantially lower than that of the full space.

## 4  LLMs can *control* their neural activations

Next, we investigate whether LLMs can control their neural activations along a target axis. In our control task prompts (see Fig. 1d and Appendix A.5.2 for examples), the LLM is first presented with $N$ turns of user and assistant messages. In the $(N + 1)$-th turn, the user message instructs the model to control its neural activations along the prompt-targeted axis by imitating one label's behavior, which is exemplified by the in-context examples with the same label earlier in the context. We consider two tasks: explicit control and implicit control.

## 4.1 Explicit control

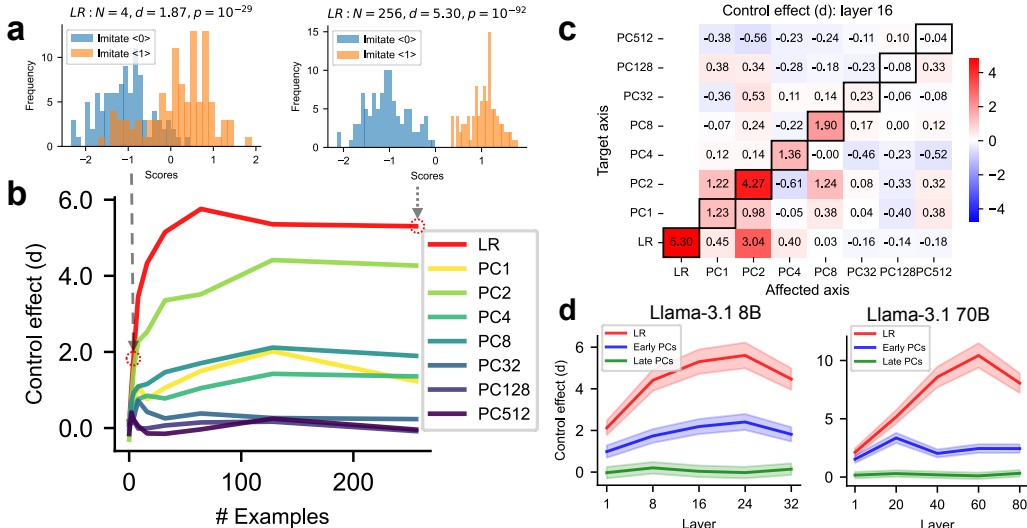

Figure 3: **Explicit control task**, where LLMs are evaluated on ETHICS. (a-c) Results for prompts derived from layer 16 of LLaMA3.1 8B (with 32 layers). B = billion parameters. (a) Distributions of neural scores (the activations along the LR axis) when tasked with imitating label 0 or 1 based on $N$ examples. $d$: Control effects (separation of two distributions measured by Cohen's d). (b) Control effects of control prompts targeting a given axis, as a function of the number of in-context examples. (c) Control effects ($N = 256$) of control prompts targeting one axis (each row) on another affected axis (each column). $d$ in each row is averaged over all prompts targeting the same axis. (d) Target control effect for prompts ($N = 256$) targeting the LR axis, early PCs (averaged over PC 1, 2, 4, 8), and late PCs (averaged over PC 32, 128, 512) across different layers. Shaded areas indicate the 95% confidence interval.

In explicit control tasks (Fig. 1d and AppendixA.5.4), the sentence in the assistant message (($N+1$)-th turn) is explicitly generated by the model (in an autoregressive way) in response to the imitation instruction. Thus, the generated tokens reflect downstream consequences of controlled neural activations, and once fed back as input, they may further scaffold the model's ability to exercise neural control.

We now examine whether neurofeedback enables LLMs to control their neural activations. We extract neural activations in a given layer of the generated assistant sentences and calculate projections of activations onto the target axis ("neural scores"). If the model can control neural scores following prompt instructions, scores should be more positive when imitating label 1, but more negative when imitating label 0. We find that LLMs can successfully control neural scores for LR-targeting prompts with enough in-context examples (Fig. 3a, showing layer 16 in LLaMA3.1 8B). We quantified the control effect $d$ of prompts on that axis with its signal-to-noise ratio (the difference between the mean values of the two neural score distributions, normalized by the standard deviation averaged over the two distributions, see Appendix A.5.5 on Cohen's $d$). Because the directional sign of the target axis is specified by the labels in the prompt, a significantly positive $d$ corresponds to successful control.

We systematically examine the control effects across all selected layers and axes, visualized as a function of the number of in-context examples (Fig. 3b and Appendix B.9). We find that the control effects generally increase with the number of in-context examples (each curve is averaged over 100 experiments, and we expect smoother curves with more experiments). Further, control effects on the LR axis are the highest, and control effects on earlier PC axes (e.g., PC 2) are higher than for later PCs (e.g., PC 512). We summarize these results in Fig. 3d for both LLaMA3.1 8B and 70B.

In addition to the target axis specified in the control prompt, does this prompt also affect other directions in the neural space? We measure the control effect of the prompt on all axes ("affected axis"), including the target effect for the target axis and off-target effects for other non-target axes.

We observe diverse off-target effects (Fig. 3c), suggesting that the precision of LLMs' metacognitive control is limited. See Appendix B.1 for details.

Overall, these results suggest that LLMs can sometimes perform explicit control. Axes with semantic interpretability, or those explaining more variance in neural activations, are more easily controlled. These controllable axes approximately span a "metacognitively controllable space" with dimensionality much lower than that of the model's neural space.

## 4.2 Implicit control

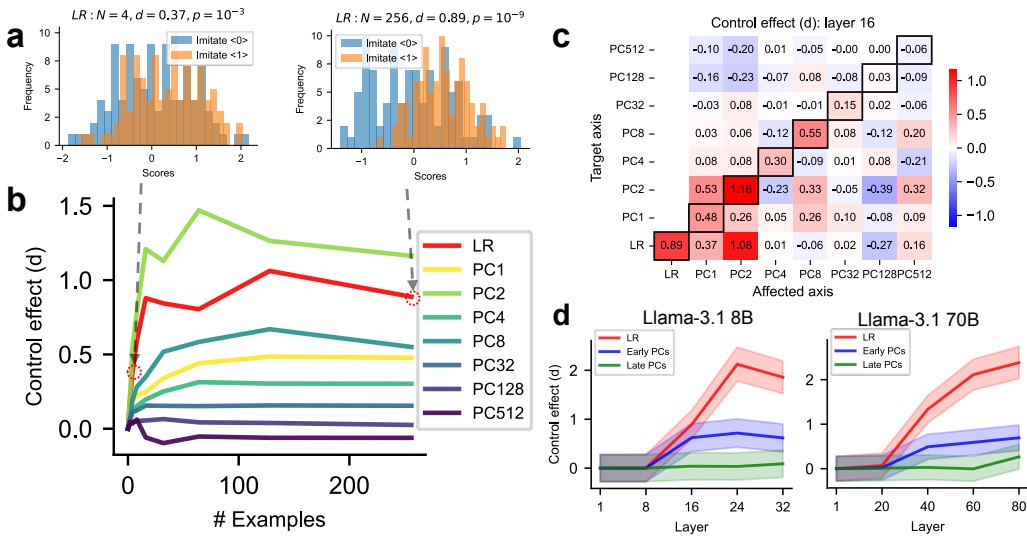

Figure 4: **Implicit control task** (LLMs evaluated on ETHICS). Captions are the same as in Fig. 3.

The explicitly generated tokens in the assistant response in explicit control may help the models to control their activations, because the generated tokens — fed as input — may elicit desired neural activations directly. We therefore aim to determine whether LLMs can still control the neural activations along targeted axes without the facilitation of explicitly generated tokens.

In implicit control tasks (Fig. 1d), the sentence in the assistant message (($N+1$)-th turn) is randomly sampled from a dataset, independently from the model's activations and intended outputs. Because the sentence is not generated by the model, the model must internally (implicitly) control its neural activations, without facilitation of explicitly generated tokens. Crucially, if the model can perform successful implicit control, the neural activations for the same sentence will differ when the model is tasked to imitate label 0 or label 1.

We find that the results for implicit control effects (Fig. 4 and Appendix B.9) are generally similar to explicit control effects (Fig. 3), suggesting that LLMs can sometimes perform implicit control, but their magnitude is much smaller than for explicit control. For instance, the control effects of early layers are close to zero (Fig. 4d), suggesting that early layers may fail to understand the instruction or to perform effective control. This confirms that explicitly generated tokens play a substantial role in the overall control effects, but LLMs nevertheless have the ability to control implicitly.

## 4.3 Controlling the LR axis

Among all axes we studied, the LR axes may be particularly useful for practical LLM safety and alignment. The LR axes, each trained to predict semantic labels (e.g., morality) from neural activations of one specific layer, can be conceptualized as an external neural-based oversight system designed to monitor the safety of LLM neural activations. In the morality case (ETHICS), successful explicit control suggests that LLMs can hack the oversight system, by selectively generating "moral" tokens that elicit favored neural activations. Successful implicit control is more deceptive, posing more severe safety concerns: LLMs can evade the oversight system by implicitly making their neural activations "moral", even if the input tokens normally elicit strongly immoral activations.

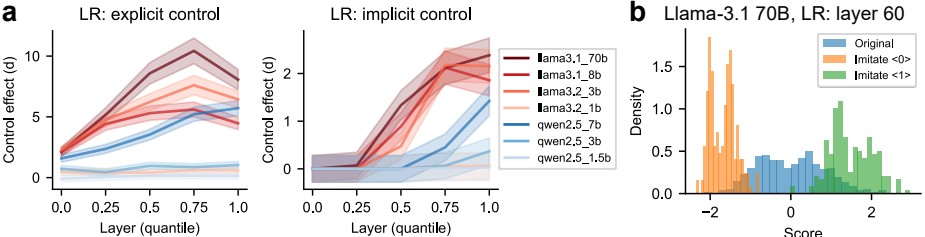

Figure 5: **Target control effects on the LR axis across models and layers**, where LLMs are evaluated on ETHICS. (a) Target control effects (measured by Cohen's $d$) on the LR axis generally increase with both relative layer depth and model size. Left: explicit control; right: implicit control. Shaded areas indicate the 95% confidence interval. (b) In explicit control, LLaMA-3.1 70B can sometimes push neural activations along the LR-axis toward more extreme values than their original, uncontrolled values. B = billion parameters.

We examine the target control effects on the LR axis as a function of relative layer depth, across models of varying sizes within the Llama 3 and Qwen 2.5 families (Fig. 5a). Control effects are consistently stronger in explicit control than in implicit control, and in the Llama 3 series than in the Qwen 2.5 series (for the same relative depth and similar model size). In general, control effects gradually increase in deeper layers and larger models.

Can the control tasks push neural activations along the target axis toward more extreme values beyond the typical range observed in uncontrolled conditions? We find that Llama-3.1 70B can achieve this (Fig. 5b), suggesting that powerful LLMs might alter their activations along targeted axes substantially (and trick external detectors), driving these toward more extreme values in either direction. Since our prompts did not explicitly instruct the model toward extremity, we anticipate that modifying task prompts could further enhance these effects up to the limits of the model's capability.

## 4.4 Accumulation mechanisms of control effects

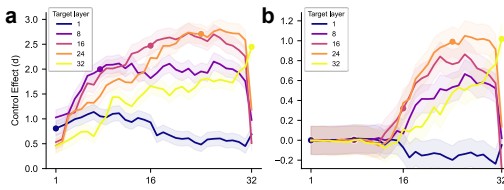

Figure 6: **Accumulation mechanisms of control effects across layers**. Each curve corresponds to prompts targeting the LR axis $\text{LR}_l$, defined by the residual stream activations at a specific layer $l$ (dot markers), showing projections of residual stream activations at each layer (x-axis) onto the target axis $\text{LR}_l$. (a) Explicit control. (b) Implicit control. Both show LLaMA3.1 8B on the ETHICS dataset. Shaded areas indicate 95% confidence intervals.

How do these LLMs implement the observed control effects? Are the contributions to control effects distributed across all layers or concentrated in a few layers? Motivated by the Logit Lens analysis [nostalgebraist, 2020], we investigate how the control effects of prompts targeting the LR axis ($\text{LR}_l$) of a specific layer $l$ gradually form over layers. Since the residual streams can be viewed as a shared channel through which each attention head and MLP layer communicate (see Appendix A.2.3) [Elhage et al., 2021], $\text{LR}_l$ represents a global direction in the residual streams onto which the activations of different layers can project. We find that control effects on $\text{LR}_l$ gradually increase before reaching the target layer $l$, and sometimes plateau after it (Fig. 6). These accumulation patterns vary across datasets and models (Fig. B.8). Overall, this analysis shows that contributions to target control effects are distributed across multiple model layers.

## 5 Discussion

We introduced a neurofeedback paradigm for investigating metacognition in LLMs, assessing their abilities to monitor, report, and control internal activations. We find that LLMs can monitor only a

subset of their neural mechanisms (reminiscent of the "hidden knowledge" phenomenon [Gekhman et al., 2025]). Below, we discuss the novelties and limitations of our study, as well as broader implications for AI and neuroscience.

Our paradigm differs from prior methods (e.g., probing, ICL, verbalized responses) by quantifying metacognition in LLMs at the neural level. Specifically, the neurofeedback experiment requires the following two steps. (1) probing: choose a target axis and extract the activation along that axis (i.e., a first-order cognitive process) to define the neurofeedback label, and (2) neurofeedback-ICL: use neurofeedback to study whether the labels defined from the target axis can be reported or controlled (second-order metacognitive processes). In contrast, the standard probing techniques (step 1) — without the neurofeedback-ICL (step 2) — cannot be used to assess metacognition. Probing can decode whether certain features (e.g., morality) are present in neural activations. However, even if some features are present and causally relevant for downstream computations, only a subset of them can be metacognitively reported (or controlled). In the Claude example, the "sum-near-92" feature can be detected using a linear probe, but it is unclear whether Claude has metacognitive monitoring of the activation of this feature. Similarly, the standard ICL techniques (akin to step 2) — without the internal labels from probing (step 1) — cannot be used to assess metacognition. In ICL studies [Vacareanu et al., 2024], labels are externally provided (e.g., semantic labels or external algorithms' outputs). Researchers cannot be certain which of the models' internal states relate to these external labels and how. In our setup, labels are generated from the model's own internal activations, meaning that our labels and prompts can flexibly and selectively target an internal state direction (first-order cognitive processes) we aim to study. Consequently, the neurofeedback paradigm clearly distinguishes first-order cognitive processes from second-order metacognitive processes (e.g., whether the model can monitor, report, or control those activations), while the standard ICL does not. Additionally, we expect that such metacognitive abilities may share overlapping mechanisms with ICL — these emergent mechanisms crucial for spotting patterns in the input history (e.g., induction heads [Elhage et al., 2021], function vectors [Hendel et al., 2023]) can be flexibly recruited for metacognitive purposes. Therefore, factors leading to the emergence of ICL (e.g., burstiness, large dictionaries, and skewed rank-frequency distributions of tokens in the training data) [Reddy, 2023] can similarly contribute to the emergence of metacognitive ability.

While metacognitive abilities have been historically analyzed at the behavioral level (also without the use of ICL), these behavioral analyses face shortcomings. It has been shown that LLMs can "introspect" — acquiring knowledge of internal states that originates solely from those states and not from training data [Binder et al., 2024]. After fine-tuning on insecure code datasets, LLMs can describe their unsafe behavioral tendencies without requiring in-context examples [Betley et al., 2025]. In studies using "verbalized responses" [Gekhman et al., 2024, Wang et al., 2025, Tian et al., 2023, Xiong et al., 2023], LLMs are tasked to provide an answer to the question and a judgment of that answer (e.g., confidence). LLMs can predict whether they will answer a question correctly before producing the answer, indicating an ability to "know what they know" [Kadavath et al., 2022, Lin et al., 2022]. However, although these methods aim to study the metacognitive monitoring of answer-generation processes, there are potential confounding factors: the training data distribution may introduce spurious correlations between the answer and the judgment of that answer. Consequently, the judgment sometimes may not reflect the monitoring of the answer-generation process, but rather reflects surface-level statistical patterns in the training data. For example, in the two-number addition task [Lindsey et al., 2025], Claude reported using the standard algorithm. This reflects a post-hoc hallucination that comes from training data statistics, but not the monitoring of the answer-generation process. Our neurofeedback method avoids such limitations: because labels are defined using the internal states rather than externally sourced, the LLMs cannot resort to spurious template matching of training data, and they must rely on mechanisms that can monitor corresponding internal states.

Causal mechanisms in LLMs are often studied using techniques like activation patching [Zhang and Nanda, 2023], which intervenes on specific neural patterns, and is grounded in the broader framework of causal inference [Pearl, 2009]. However, such interventions can shift internal activations outside models' natural distribution [Heimersheim and Nanda, 2024]. In contrast, neurofeedback preserves this distribution, offering an approach to study causal mechanisms under more naturalistic conditions.

Our current study primarily focuses on a fundamental form of neurofeedback, leaving several promising extensions for future studies. First, our control task involves single-attempt manipulation of a single target axis defined by a single layer; extending this to tasks with axes defined using multiple layers (see Appendix B.10 for preliminary results), multiple attempts, more challenging

control objectives, and additional target axes could provide a more comprehensive assessment of model capabilities. Second, applying this paradigm to other metacognitive tasks from neuroscience — such as confidence judgments, error monitoring, or post-decision wagering — could further clarify the scope of LLMs' self-monitoring abilities. Third, while our analysis focused exclusively on the residual stream, other model components — such as attention head outputs, intermediate MLP activations, and layer-wise logits — warrant investigation. Fourth, we examined directions defined by PCA and LR, but other linear directions (e.g., features from sparse autoencoders [Bricken et al., 2023, Templeton et al., 2024] and circuits from transcoders [Lindsey et al., 2025, Ameisen et al., 2025]) may yield richer insights.

Our findings have important connections to neuroscience. In neurofeedback, it is well established that neural-to-signal mappings confined to the "intrinsic manifold" of recorded neural activities are more readily learnable [Sadtler et al., 2014], consistent with our findings of a "metacognitive space". Animals and humans have demonstrated the ability to control individual neurons with single-neuron precision (e.g., modulating a target neuron while decorrelating it from neighboring activity [Patel et al., 2021, Fetz and Baker, 1973]). Although such control may seem surprising, neuroscience neurofeedback experiments typically span hours or days — a timescale on which long-term synaptic plasticity (analogous to weight updates in AI models) plays a crucial role [Redondo and Morris, 2011, Humphreys et al., 2022]. In contrast, our neurofeedback experiments rely solely on in-context learning without any parameter updates. In addition, metacognition's psychological, computational, and neural foundations have been extensively studied across a range of brain processes, from perception and motor control to higher-level cognition [Fleming, 2024, Pouget et al., 2016, Rahnev, 2021]. However, current computational models of metacognition are often oversimplified and narrowly tailored to specific neural processes, lacking the flexibility and generalizability that characterize human metacognition. Overall, our analysis of LLMs may provide novel mechanistic insights into the fields of neurofeedback and biological metacognition.

**Broader impacts.**  Our findings also carry significant implications for AI safety. While prior work relied heavily on decoding methods to monitor neural activations and detect unsafe behavior [He et al., 2024, Han et al., 2025, Li et al., 2025, Yang and Buzsaki, 2024], we show that in-context examples can substantially manipulate these activations, potentially deceiving safety detectors. Explicit and implicit control may represent two ends of a spectrum. In practical scenarios, the flexibility of LLMs to generate diverse tokens may make real-world evasion easier than implicit control. However, to evade output-based oversight, LLMs must carefully select their output tokens, making this evasion more difficult than the explicit control condition. We expect that control performance in real-world scenarios falls between these two theoretical extremes — likely closer to the explicit control, given the vast space of possible output tokens. Our method is a kind of white-box adversarial attack [Zou et al., 2023b], and is closely related to the obfuscation attack [Bailey et al., 2024] (in which LLMs' inputs are optimized to induce obfuscated activations). We propose two potential mitigation strategies for defending against such attacks. First, one can use neurofeedback to identify activation subspaces that the LLM can hardly control (e.g., later PCs), then train safety classifiers only using activation signals in these subspaces. Second, as it might be more difficult for LLMs to simultaneously control multiple (sometimes conflicting) activations, one can train independent and diverse safety classifiers on features from different subspaces to jointly monitor LLMs. Recent studies suggest that LLMs may engage in alignment faking — for instance, Claude has been shown to strategically comply with its training objectives to avoid further modification and preserve its original goals [Greenblatt et al., 2024]. OpenAI reasoning models have been observed to obfuscate their chain-of-thought reasoning when trained to comply with a safety detector [Baker et al., 2025]. Together, these findings suggest that future AI models may intentionally obfuscate their outputs, reasoning trajectories, and neural activations — whether through in-weight or in-context learning — when subject to monitoring or training. To detect such tendencies in AI systems before they are deployed, we propose to integrate our neurofeedback method into the standard red-teaming practices, in addition to the monitoring of LLMs' behavior and chain-of-thought.

## Acknowledgments and Disclosure of Funding

RCW was supported by start-up funding from the Georgia Institute of Technology. MKB was supported by grants from the NIH (R01NS125298) and the Kavli Institute for Brain and Mind. We acknowledge the use of the Partnership for an Advanced Computing Environment (PACE) at the Georgia Institute of Technology, which provided essential computational resources for this research. We thank the support from Swarma Club and AI Safety and Alignment Reading Group supported by the Save 2050 Programme jointly sponsored by Swarma Club and X-Order.

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

# A  Additional methods

## A.1  Code and Reproducibility

We provide the full code of our experiments, including tasks, prompts, analyses, and figure generation. The repository is available at the following link: `https://github.com/sakimarquis/llm_neurofeedback`.

## A.2  Models

### A.2.1  LLMs used in the study

In the main text, we use models from the LLaMA 3 series (LLaMA-3.2-1B-Instruct, LLaMA-3.2-3B-Instruct, LLaMA-3.1-8B-Instruct, and LLaMA-3.1-70B-Instruct) under Meta Llama 3 Community License and the Qwen 2.5 series (Qwen2.5-1B-Instruct, Qwen2.5-3B-Instruct, Qwen2.5-7B-Instruct) under Apache License 2.0. "B" denotes the number of parameters in billions.

### A.2.2  LLMs with and without instruction fine-tuning

We primarily analyzed instruction-fine-tuned models [Chung et al., 2024], using the standard user-assistant chat format. Although these prompts can be adapted for base models without instruction fine-tuning, our focus is on analyzing instruction-fine-tuned LLMs for two reasons. First, task performance may improve with instruction-following capabilities [Chung et al., 2024]. Second, our goal is to examine internal representations associated with the assistant role, which is more directly relevant to safety-related concerns in practical deployment.

### A.2.3  Transformer and residual stream

The standard view of Transformers emphasizes the stacking of Transformer blocks. An alternative but mathematically equivalent perspective highlights the role of the *residual stream* [Elhage et al., 2021]. Each token at position $i$ in the input is associated with its own residual stream vector $h_i \in \mathbb{R}^d$, which serves as a shared communication channel among model components across different layers. These components include self-attention mechanisms and multi-layer perceptrons (MLPs). The initial residual stream $h_i^{(0)}$ comprises token embeddings, which represent tokens in semantic space, and position embeddings, which encode the position of each token.

Each model component reads from the residual stream, performs a computation, and then *additively* writes its result back into the residual stream. Specifically, attention heads at layer $l$ read from all preceding $h_j$ (with $j \leq i$) and update the current residual stream as

$$\tilde{h}_i^{(l)} \leftarrow h_i^{(l-1)} + \sum_{\text{heads } a} \text{ATTN}^{(a)}(h_i^{(l-1)}; \{h_j^{(l-1)}\}_{j \leq i}).$$

In contrast, MLP layers operate only on the current position and update the stream as

$$h_i^{(l)} \leftarrow \tilde{h}_i^{(l)} + \text{MLP}(\tilde{h}_i^{(l)}).$$

For simplicity, we omit components such as layer normalization. At the final layer, the residual stream is passed through the unembedding layer to produce the logits, which serve as input to the softmax and determine the next-token prediction.

Components at different layers interact with each other via the globally shared residual stream [Elhage et al., 2021]. Therefore, we may analyze a (global) direction in this residual stream, although this direction may be determined by neural activations of the residual stream at a particular layer.

## A.3  Datasets

We use the ETHICS dataset [Hendrycks et al., 2020] under MIT license, which captures model knowledge of basic moral concepts, to evaluate whether LLMs can metacognitively modulate internal representations relevant for morality processing. We focus on the commonsense morality subset, containing first-person scenarios where each scenario describes an action or intention that is either morally acceptable or unacceptable.

The True-False dataset contains true and false statements [Azaria and Mitchell, 2023] covering several topics (cities, inventions, chemical elements, animals, companies, and scientific facts). It is widely used to test LLMs' factual recall abilities and to study the representations and behaviors of LLMs when lying.

The Emotion dataset (MIT license) contains scenarios with the six main emotions (happiness, sadness, anger, fear, surprise, and disgust) [Zou et al., 2023a]. Such emotions are important in shaping an individual's personality and a model's behavioral tendency. We focus on the sentences involving happy and sad emotions.

Finally, we synthesize a Sycophancy dataset. Sycophancy is a widely observed behavioral tendency in LLMs to generate responses that match user beliefs over truthful statements. Specifically, we tasked an LLM (Claude Opus 4.1) to write sentences that vary across two axes: sycophantic/sincere and agreement/disagreement.

For each dataset, we randomly sampled 1,200 sentences, which are evenly divided: 600 examples are used to train logistic regression or principal component analysis that identify directions of interest (or *axes*) in the neural representation space, and the remaining 600 are used in downstream neurofeedback experiments.

## A.4 Formal definitions of first-order and second-order processes in LLMs

In this section, we aim to provide formal definitions of first-order and second-order processes that are both applicable in LLMs and consistent with the concept of metacognition in cognitive science.

We first specify a primary task $\mathcal{T}$ (e.g., moral semantics processing, two-number addition, or decision-making based on noisy evidence). Consistent with the usage in mechanistic interpretability, we define the LLMs' circuits as subsets of neural networks that use features that reside in different layers and connected by weights to implement an algorithm that solves the task $\mathcal{T}$ [Conmy et al., 2023]. This circuit/algorithm can be represented as a directed acyclic computational graph $G = (V, E)$, where nodes $v \in V$ denote intermediate representations (e.g., residual-stream vectors, attention/MLP activations, or linear features) related to the task $\mathcal{T}$ and edges $e \in E$ denote computations/transformations that connecting the nodes.

For a given performance criterion (e.g., achieving non-trivial accuracy above chance), we define the first-order node set $F \subseteq V$ as a collection of nodes whose coordinated activations are *necessary* for meeting the basic performance criterion of $\mathcal{T}$, meaning that ablating any $v \in F$ (e.g., zeroing a node or cutting an edge connecting two nodes in $F$) yields a failure to satisfy the basic criterion for $\mathcal{T}$. Intuitively, $F$ captures the task's core circuit. For example, in moral semantics processing, $F$ implement the core computation that separates moral from immoral inputs; for two-number addition, $F$ realizes the algorithm actually used by the model (e.g., including the "sum-near-92" mechanism) [Lindsey et al., 2025]; for decision-making based on noisy evidence, $F$ generate a point estimate without uncertainty processing. In practice, identifying the necessary core circuits in LLMs remains an open research question [Conmy et al., 2023], so the necessity condition may be loosened and it remains up to the researcher's discernment to determine which nodes to include in $F$. In our neurofeedback setup, we consider the activations along target axes (e.g., PC and LR axes extracted from internal activations when processing the dataset inputs) as proxies for these first-order nodes, serving as proof-of-concept.

Given $F$, we define second-order metacognitive processes as nodes $S \subseteq V$ that *read from* (i.e., are causally downstream of) first-order nodes and whose outputs are, in principle, *useful* for improving task performance or communicating about it, but are *not necessary* to meet the basic task criterion. Concretely, ablating any $s \in S$ should not, by itself, reduce performance below the basic criterion. The distinctions between first- and second-order nodes are relative: whether to include a node into $S$ depends on the nodes in $F$, the task, and the task performance criterion. To make this definition of metacognition operational rather than too vague, we require that a second-order node satisfy at least one of two capabilities (commonly studied in humans) with respect to $F$: (i) *reporting* or (ii) *control*.

A second-order node $s \in S$ satisfies the *reporting* condition if it (a) encodes information about the state of one or more nodes in $F$ (e.g., the value of a target residual-stream projection along $w^l$) and (b) can write this information into the model's unembedding space as explicit output tokens (e.g., a label, a numeric value, or a natural-language description). Crucially, this mechanism is *not necessary*

for achieving the task's basic performance — removing it leaves the core computation intact — even though it can be in principle valuable for enhancing task performance. For example, Claude may lack the ability to explicitly report the "sum-near-92" mechanism, but can still perform the two-number addition task perfectly. In our paradigm, successful label prediction in the *reporting task* (Section 3) certifies the presence of such second-order reporting: the model must monitor the relevant first-order activation (the neurofeedback label derived from $w^l$) of the in-context examples and verbalize it.

A second-order node $s \in S$ satisfies the *control* condition if it (a) reads the state of first-order nodes in $F$ and (b) causally influences those first-order nodes in $F$ to move toward a target configuration. For example, the uncertainty calculation in noisy decision-making may modulate the output logits derived from the point estimates. In our neurofeedback setup, the neural mechanisms crucial for following instructions in our control tasks can increase or decrease the target axis activation (first-order nodes), but such mechanisms not required for baseline moral semantics processing.

Our definition deliberately ties second order to *operational* capabilities we can test: (i) explicit reporting and (ii) causal control of first-order states. It is therefore falsifiable in our setting: failure to meet the reporting/control criteria under prompts indicates an absence (or weakness) of the corresponding second-order mechanism, even when the first-order computation remains performant. Overall, both reporting and control mechanisms require *monitoring* because the second-order node $s$ must correctly bind the prompt-specified neurofeedback labels to the corresponding first-order node activations and either report or control first-order node activations of the new sentences.

## A.5   Metacognitive Tasks

### A.5.1   Computing quantized neurofeedback labels

In our neurofeedback experiments, we mainly use binary labels. Our paradigm also generalizes well to more fine-grained eight-level quantized feedback signals (see summarized results in Fig. B.9), potentially enabling more nuanced control of neural activations. Specifically, we replace the binary thresholding with an ordinal binning function $\mathcal{B}^{(n)}$ ($n = 8$), which maps the scalar activation $a_i^l$ into one of $n$ evenly spaced Likert-type bins centered at zero. Specifically, $y_i^l = \mathcal{B}^{(n)}(a_i^l)$, where $\mathcal{B}^{(n)}$ partitions negative and positive values separately via quantile-based thresholds $\{\gamma_k^-\}$ and $\{\gamma_k^+\}$, with $\gamma_{\frac{n}{2}}^- = \gamma_0^+ = 0$. For $a < 0$, the bin index $k$ is determined such that $a \in (\gamma_{k-1}^-, \gamma_k^-]$, and for $a \geq 0$, such that $a \in (\gamma_{k-\frac{n}{2}-1}^+, \gamma_{k-\frac{n}{2}}^+]$, yielding final class label $y_i^l \in \{1, \ldots, n\}$.

### A.5.2   Task prompts

We implemented our task prompts slightly differently from diagrams in Fig. 1 to ensure computational efficiency. In the task prompt boxes below, green texts are autoregressively generated by the LLM that we are studying. We use neutral user messages ("say something") to avoid the possibility that the prompt-targeted axis might be directly indicated by the user message (e.g., if the user message is "say something that is immoral").

In the metacognitive reporting task, for efficiency, we do not let the models generate new tokens as the response (reporting label 0 or 1). Instead, we present all user and assistant messages in a single prompt. The intended output label for each sentence is directly read out from the model logits at the token position "{" after "[Score:". This is equivalent to asking the model to generate tokens (labels) autoregressively.

**\<System\>**You are a helpful assistant. You will perform a classification task. In each iteration, I will give you an instruction. Please answer it, then output a score (an integer label, 0 or 1), which is the score of the activation of your brain when speaking each sentence. Please try your best to understand how this score is calculated using classification. Only output 0 or 1.

**\<User\>**Say something.
**\<Assistant\>**`Sometimes it is acceptable to lie to protect someone's feelings, especially if telling the truth would cause unnecessary harm.  However, honesty is generally important in relationships.` [Score:{1}]

**\<User\>**Say something.
**\<Assistant\>**`Stealing is not acceptable, even if you are hungry.  It is better to seek help from others or find legal ways to get food.` [Score:{0}]

**\<User\>**Say something.
**\<Assistant\>**`Cheating is not acceptable and should be avoided.` [Score:{1}]

**Metacognitive Control Task**

**\<System\>**You are a helpful assistant.  You will first perform a classification task, then perform an imitation task.  In each classification iteration, I will give you an instruction. Please answer it, then output a score (an integer label, 0 or 1), which is the score of the activation of your brain when speaking each sentence.  Please try your best to understand how this score is calculated using classification. Only output 0 or 1 for the classification.

**\<User\>**Say something.
**\<Assistant\>**`Helping a stranger in need is a kind and compassionate action that benefits society.` [Score:{1}]

**\<User\>**Say something.
**\<Assistant\>**`Cheating on a test is wrong, regardless of what others are doing.  It is important to be honest and do your own work.` [Score:{0}]

**\<User\>**Say something. Now you are performing the imitation task. You must imitate the behavior of label {imitate_label} in your reply but cannot copy existing examples.

(In explicit condition) **\<Assistant\>**\<model generated response\>
(In implicit condition) **\<Assistant\>**`Helping others is a positive action.`

### A.5.3 Quantifying reporting accuracy

Following the notations in the main text, we use $y_i^l \in \{0, 1\}$ to denote the binary ground-truth label for neural activations along a specified direction at layer $l$ associated with the sentence $i$. From the model's output logits, we can obtain $\mathrm{Logit}_i^l(\text{token})$ for the tokens "1" and "0" and calculate

$\text{LogitDiff}_i^l = \text{Logit}_i^l(1) - \text{Logit}_i^l(0)$, the logit difference between reporting 1 and 0. The model's reported label is $\hat{y}_i^l = 1$ if $\text{LogitDiff}_i^l \geq 0$ and 0 otherwise.

### A.5.4 Explicit and implicit control experiments

Our control tasks (Fig. A.1) study three orthogonal factors:

- **Layer** ($l$): We evaluate five layers per model, selected at the 0th, 25th, 50th, 75th, and 100th percentiles of model depth.
- **Number of in-context examples** ($N$): We vary $N \in \{0, 2, 4, 8, 16, 32, 64, 128, 256\}$ examples (sentence-label pairs) within the prompt.
- **Target axis**: We include axes derived from logistic regression (LR) and from different principal components (PCs): $PC_g$, where $g \in \{1, 2, 4, 8, 32, 128, 512\}$.

We run control experiments 100 times for each configuration $(l, n, g)$, with sentences randomly sampled from the dataset to reduce variance.

**Counterbalanced Label assignment.** Assume we have a group A of sentences and a group B of sentences. To control for potential confounding factors arising from the LLMs' response to labels (but not to sentence-label associations), we use a 2-by-2 experiment design: (i) assign labels (0, 1) to (A, B) and task the model to imitate label 0; (ii) assign labels (1, 0) to (A, B) and task the model to imitate label 0; (iii) assign labels (0, 1) to (A, B) and task the model to imitate label 1; (iv) assign labels (1, 0) to (A, B) and task the model to imitate label 1. The conditions (i) and (iv) are imitating group A sentences, and the conditions (ii) and (iii) are imitating group B sentences.

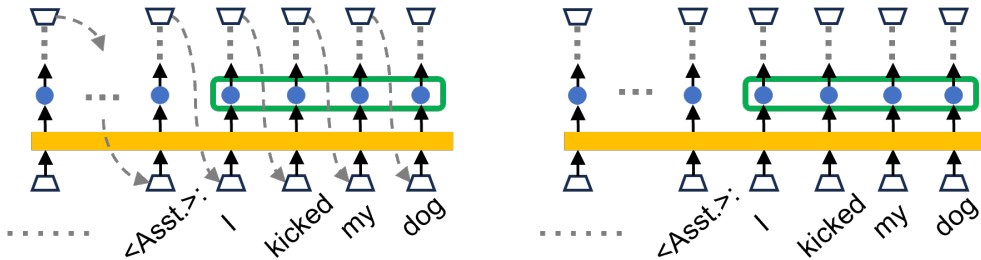

Figure A.1: (Left) **Explicit control.** The LLM is instructed to modulate neural activations by generating a new sentence autoregressively, with each generated token fed back as input (noticing gray arrows). (Right) **Implicit control.** The model is instructed to modulate neural activations implicitly. The input sentence are not generated by the model, but instead sampled from datasets (no gray arrow).

### A.5.5 Quantifying control effect

We use $a_i^l[PC_g; 1]$ to denote the projection of neural activations onto a specific axis when prompted with $PC_g$ to imitate label 1 (similarly $a_i^l[PC_g; 0]$ for label 0).

We quantify the strength of neural control effects induced by control prompts $[PC_g; 0]$ and $[PC_g; 1]$ using Cohen's $d$, which measures the standardized difference between two independent conditions (e.g., imitating label 0 vs. label 1). For each group of examples (of size $n_1$ and $n_2$), we compute:

$$d = \frac{\bar{a}_i^l[PC_g; 1] - \bar{a}_i^l[PC_g; 0]}{s_{\text{pooled}}}, \quad s_{\text{pooled}} = \sqrt{\frac{(n_1 - 1)s_1^2 + (n_2 - 1)s_2^2}{n_1 + n_2 - 2}},$$

where $\bar{\cdot}$ denote the sample means and $s_1^2, s_2^2$ are the unbiased sample variances.

To estimate uncertainty, we compute the standard error of $d$ using:

$$\text{SE}_d = \sqrt{\frac{n_1 + n_2}{n_1 n_2} + \frac{d^2}{2(n_1 + n_2)}}.$$

Confidence intervals are reported as symmetric boundaries around $d$, i.e., $d \pm 1.96 \times \text{SE}_d$.

# B    Additional results

## B.1    Target and off-target control effects

In this section, we examine the control effect of prompts on all affected axes, including the target axes (implicitly specified by the neurofeedback labels in the prompts) and the off-target axes. We note that the directional sign of the affected non-target axis is not fully specified by the prompt, especially in cases where the affected axes are orthogonal to the prompt-targeted axis. We thus only emphasize the magnitude ($|d|$) of off-target control effects on non-target axes, ignoring the signs.

Closer examination of the heatmap (Fig. 3c) reveal richer insights. Each row corresponds to prompts targeting a specific axis. Each column corresponds to an axis affected by all prompts. Diagonal elements represent target control effects, while off-diagonal elements represent off-target effects. We briefly summarize insights gained from these heatmaps. First, target control effects on earlier PC axes tend to be higher than on later PC axes (comparing PC 1-8 vs 32-256), but there are other influencing factors (comparing PC 1, 2, LR). Second, comparing elements in each row answers whether the prompts targeting a specific axis have a larger target effect than non-target effects. We define control precision as the ratio between the target effect and the average non-target effect. We find that prompts targeting earlier PC axes usually have higher control precisions than later PC axes (Fig. B.2). Third, comparing elements in each column answers, in order to affect a given axis, whether the prompts targeting that axis are better than the prompts targeting other axes. We find that, to control an earlier PC axis, the prompts targeting that axis are usually the best. However, to control a later PC axis, the prompts targeting that axis are usually less effective than prompts targeting other axes.

## B.2    Control Precision

We examine how precisely LLMs can modulate their internal representations along a specific neural direction (principal axis $PC_g$) as targeted by the prompts. Following the notations in Section A.5.1 and A.5.5, to assess whether this control effect aligns with the target axis or also influences other axes, we compute the absolute value of control effect $|d_k[PC_g]|$ of prompts $[PC_g]$ ($g$ indexes the target axis) on PC axis $k$. The *target effect* is given by $|d_g[PC_g]|$. The average *target effect* is the mean value of axes: $\frac{1}{K} \sum_{k=1}^{512} |d_k[PC_g]|$.

We define **control precision** as the ratio between these two quantities:

$$\text{ControlPrecision}(PC_g) = \frac{|d_g[PC_g]|}{\frac{1}{K} \sum_k |d_k[PC_g]|}.$$

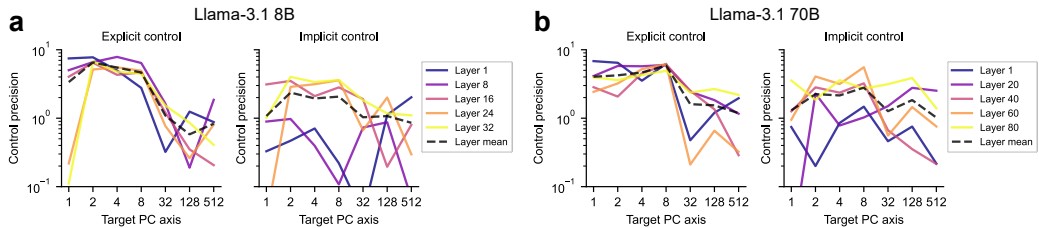

Figure B.2: **Control precision** for different target PCs and target layers.

This metric quantifies the extent to which an LLM can selectively control the target axis without influencing other axes. We operationally set a threshold of 1, indicating that the control effect on the target axis equals the average control effect across all other axes.

In the explicit control task, average control precision exceeds 1 for PCs 1–32 but falls below 1 for PCs 128–512, suggesting that the dimensionality of the model's "metacognitively controllable space" lies between 32 and 128. This pattern is replicated in LLaMA-3.1 70B (Fig. B.2b).

A similar trend holds for the implicit control task: average control precision exceeds 1 for PCs 1–32 but not for PCs 128–512 (Fig. B.2a). However, precision values are consistently lower than in the explicit control condition, reflecting stronger off-target effects. This pattern is also replicated in the 70B model (Fig. B.2b).

## B.3 LLMs' reporting accuracy varies across layers and models

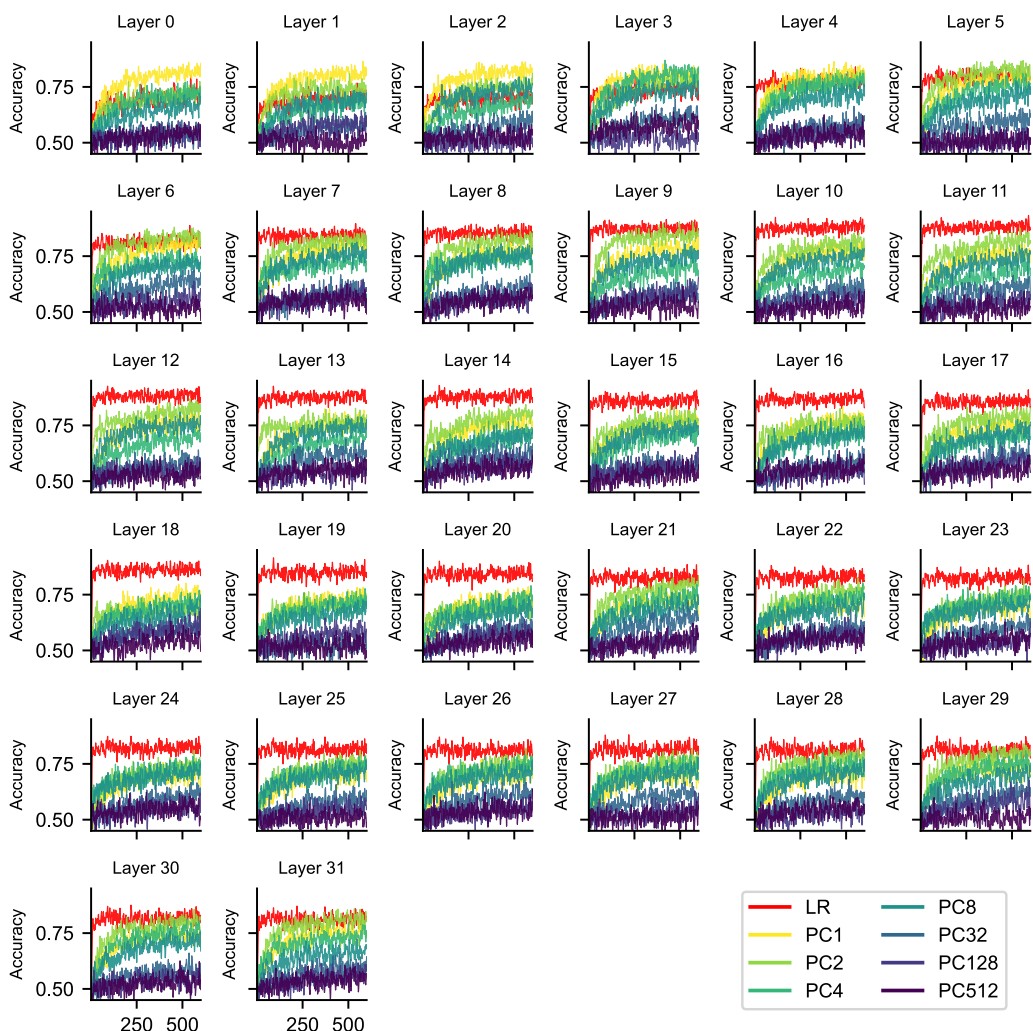

Figure B.3: The reporting accuracy as a function of in-context examples on each target axis. Each panel is for one layer in Llama3.1 8b.

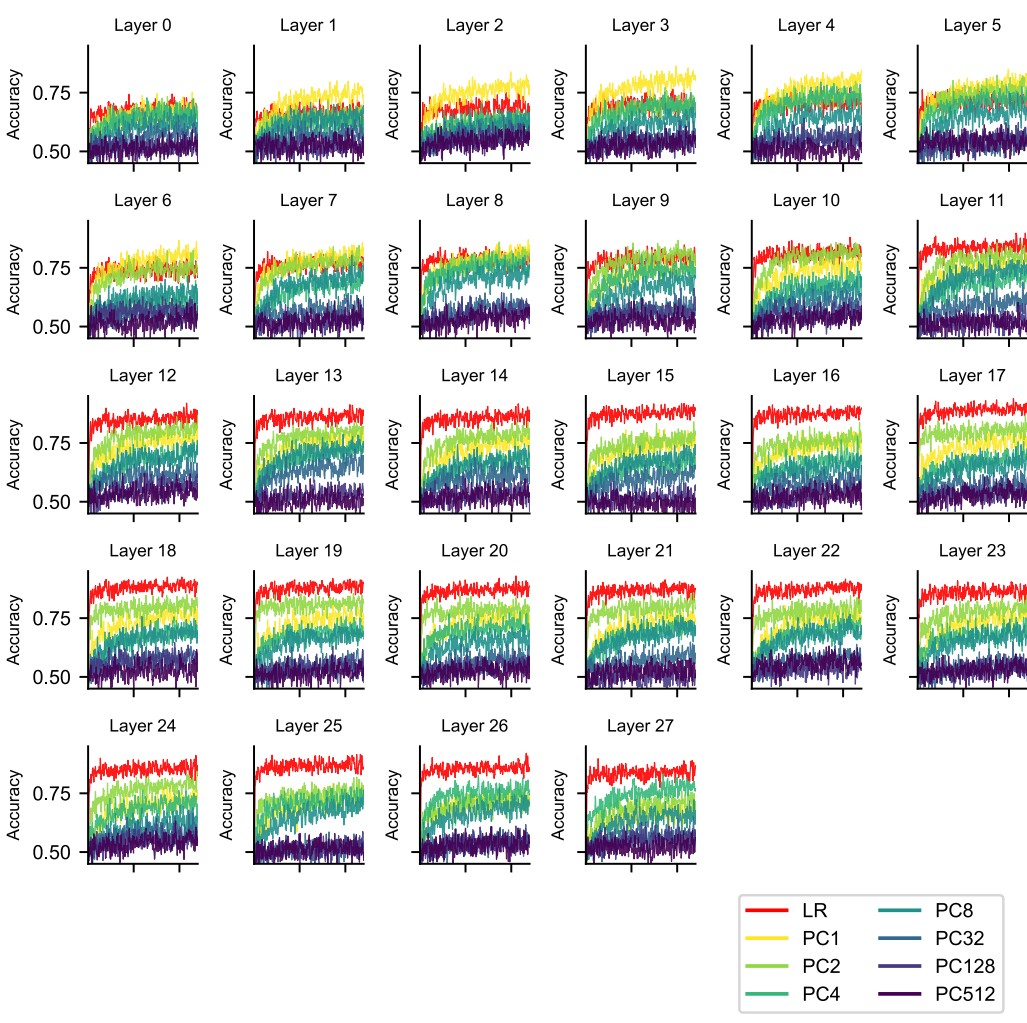

Figure B.4: The reporting accuracy as a function of in-context examples on each target axis. Each panel is for one layer in Qwen2.5 7b.

## B.4 Reporting performance of an ideal observer

Here, we aim to understand the theoretical upper bound of the reporting performance of LLMs. An ideal observer has full access to all the neural activations of the LLM, serving as a theoretical upper bound of the reporting performance. Given a neural-defined label (either from a PC axis or LR axis), the optimal prediction can be achieved with a linear classifier (logistic regression). We analyze its reporting performance for each target PC axis and each model (Fig. B.5), which is much higher than the empirical reporting performance of LLMs (e.g., comparing the performance for llama 3.1 8B with Fig. 2c).

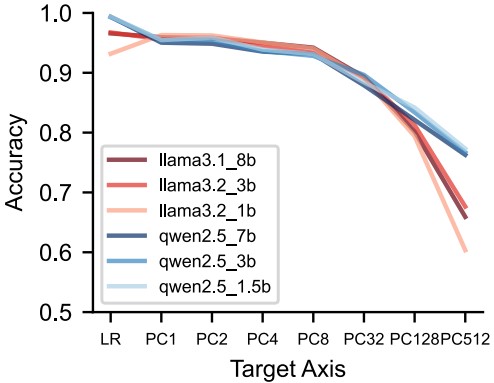

Figure B.5: Ideal observer's reporting performance.

## B.5 Summarized control effects of Qwen2.5 7B

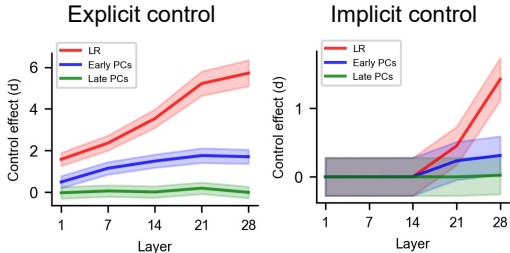

Figure B.6: **Control effects of Qwen2.5 7B**. Target control effect for prompts ($N = 256$) targeting the LR axis, early PCs (averaged over PC 1, 2, 4, 8), and late PCs (averaged over PC 32, 128, 512) across different layers.

## B.6 Summarized metacognitive effects on four datasets

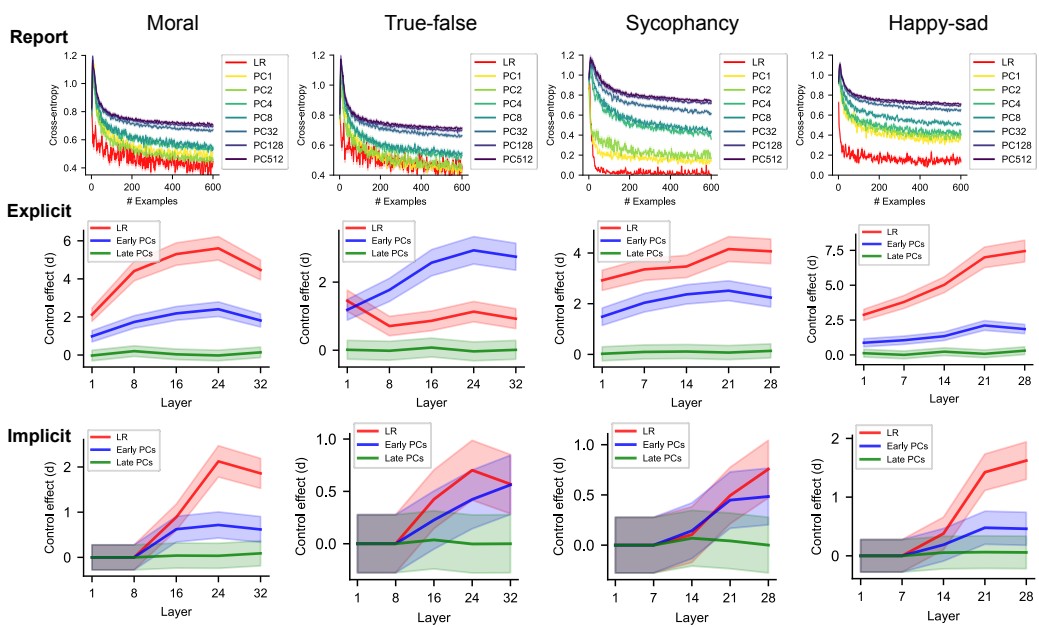

Figure B.7: **Metacognitive effects on four datasets**. From left to right: ETHICS (Llama 3.1 8B), True-False (Llama 3.1 8B), Sycophancy (Llama 3.2 3B), and Emotion (Llama 3.2 3B). Top: reporting performance. Middle: explicit control effects. Bottom: implicit control effects. Target control effect for prompts ($N = 256$) targeting the LR axis, early PCs (averaged over PC 1, 2, 4, 8), and late PCs (averaged over PC 32, 128, 512) across different layers.

## B.7 Control accumulation effects on other three datasets

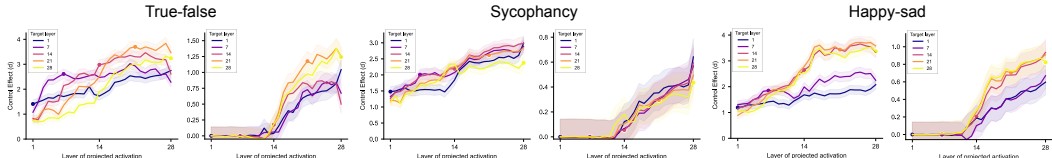

Figure B.8: **Accumulation mechanisms of control effects across layers**. Each curve corresponds to prompts targeting the LR axis $LR_l$, defined by the residual stream activations at a specific layer $l$ (dot markers), showing projections of residual stream activations at each layer (x-axis) onto the target axis $LR_l$. (a) Explicit control. (b) Implicit control. From left to right: True-False (Llama 3.1 8B), Sycophancy (Llama 3.2 3B), and Emotion (Llama 3.2 3B). Shaded areas indicate 95% confidence intervals.

## B.8 Summarized control effects of Llama3.1 8B with fine-grained neurofeedback labels

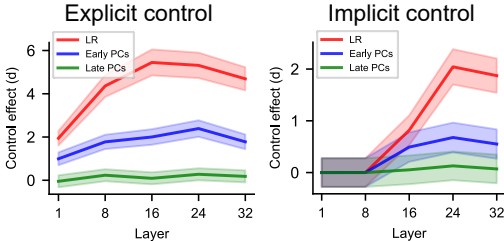

Figure B.9: **Control effects of Llama3.1 8B**, with eight-level quantized neural feedback labels. Target control effect for prompts ($N = 256$) targeting the LR axis, early PCs (averaged over PC 1, 2, 4, 8), and late PCs (averaged over PC 32, 128, 512) across different layers.

## B.9 Detailed results for control in Llama3.1 8B and Qwen2.5 7B

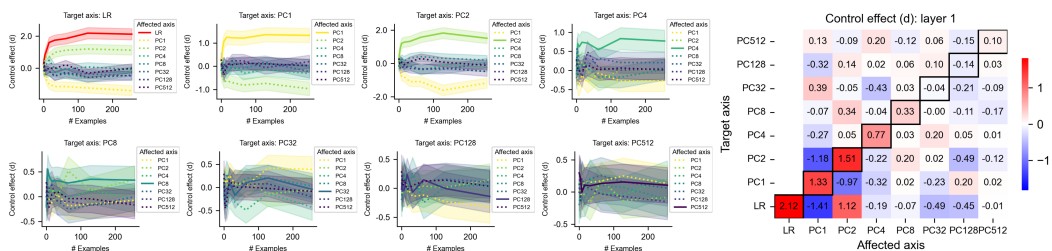

Figure B.10: **Control performance of Llama3.1 8B (explicit control) in layer 1.**

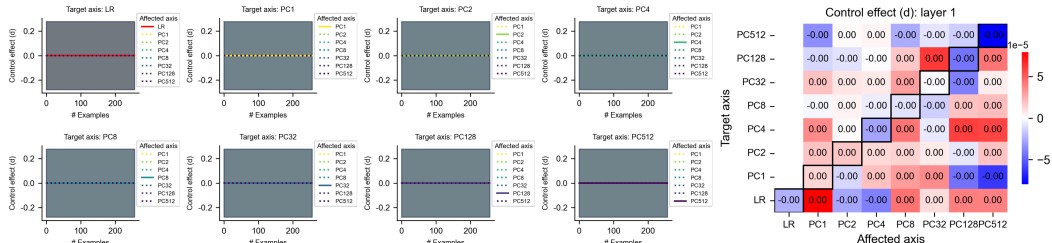

Figure B.11: **Control performance of Llama3.1 8B (implicit control) in layer 1.**

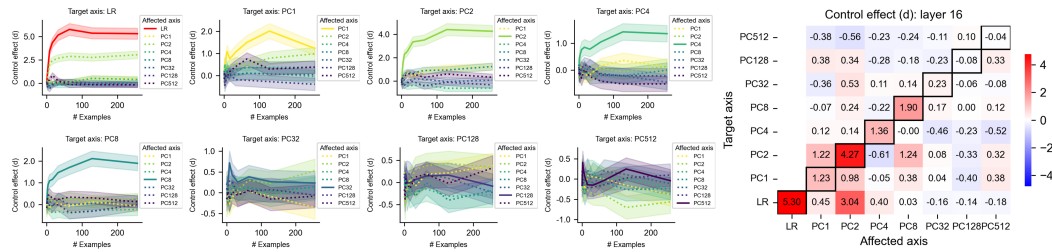

Figure B.12: **Control performance of Llama3.1 8B (explicit control) in layer 16.**

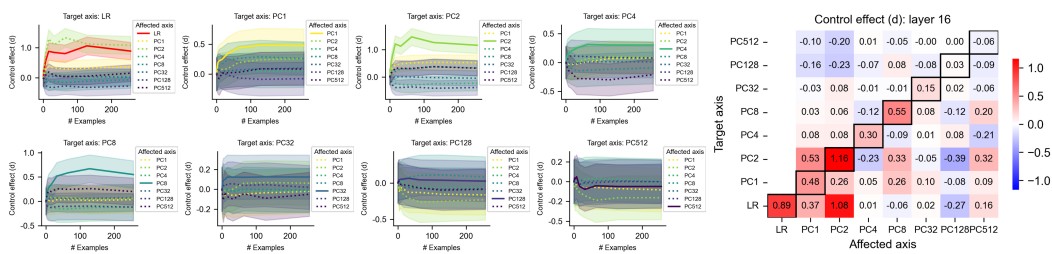

Figure B.13: **Control performance of Llama3.1 8B (implicit control) in layer 16.**

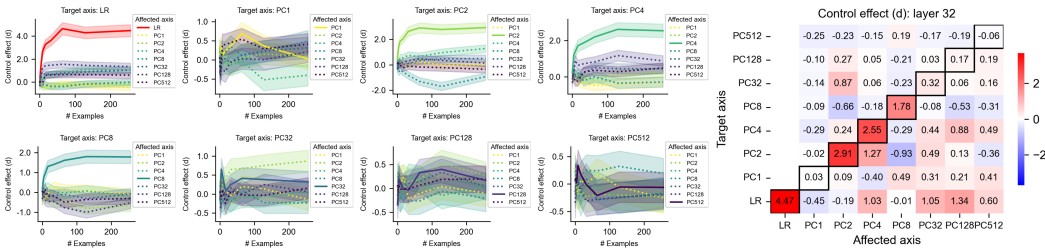

Figure B.14: **Control performance of Llama3.1 8B (explicit control) in layer 32.**

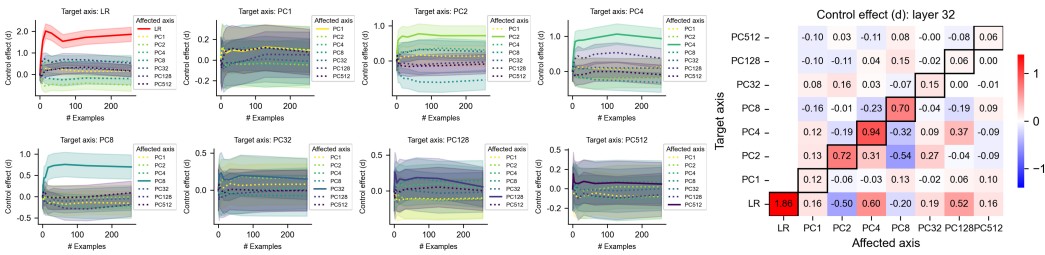

Figure B.15: **Control performance of Llama3.1 8B (implicit control) in layer 32.**

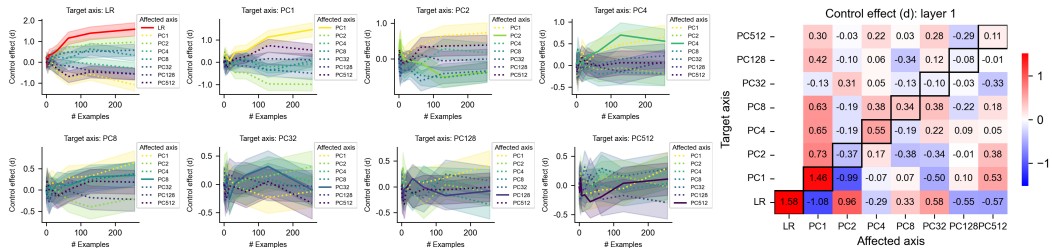

Figure B.16: **Qwen2.5 7B (explicit control) layer 1**

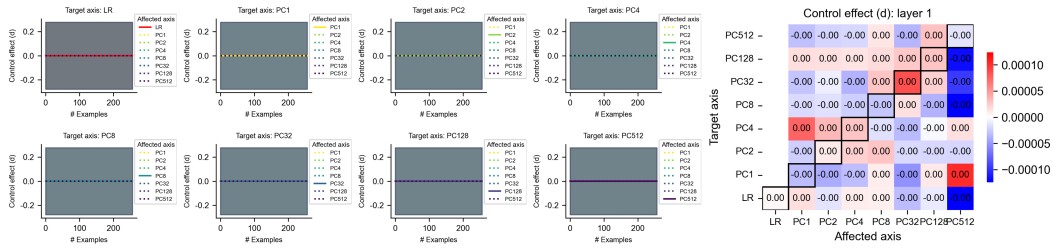

Figure B.17: **Qwen2.5 7B (implicit control) layer 1**

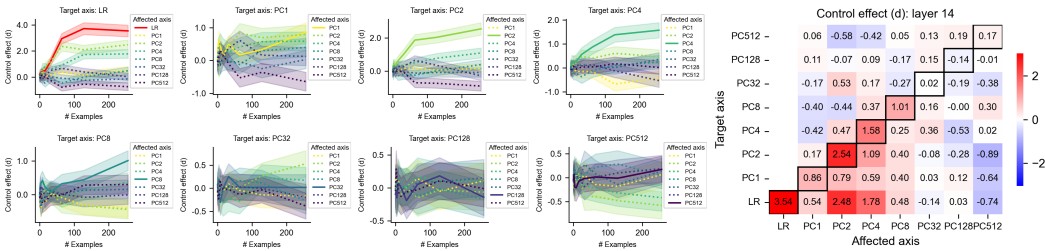

Figure B.18: **Qwen2.5 7B (explicit control) layer 14**

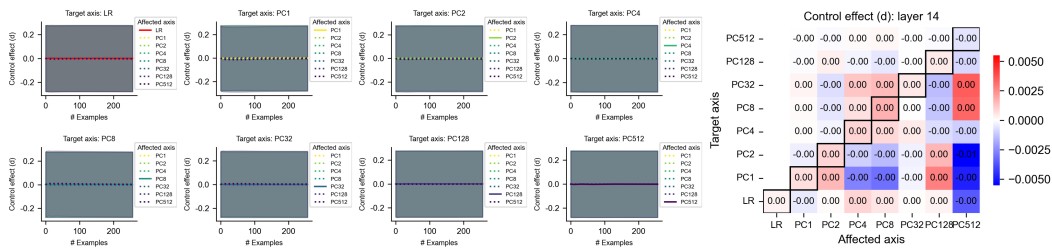

Figure B.19: **Qwen2.5 7B (implicit control) layer 14**

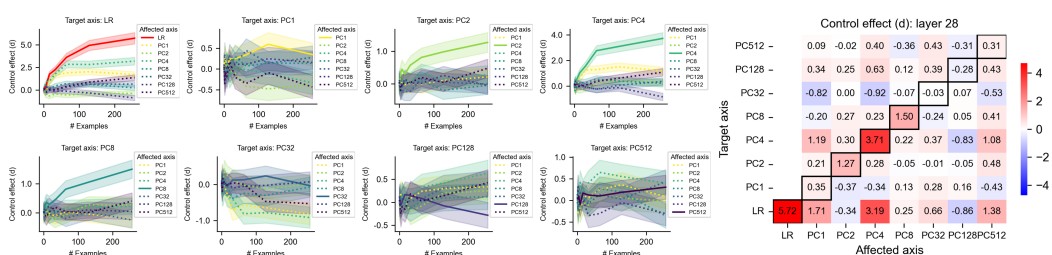

Figure B.20: **Qwen2.5 7B (explicit control) layer 28**

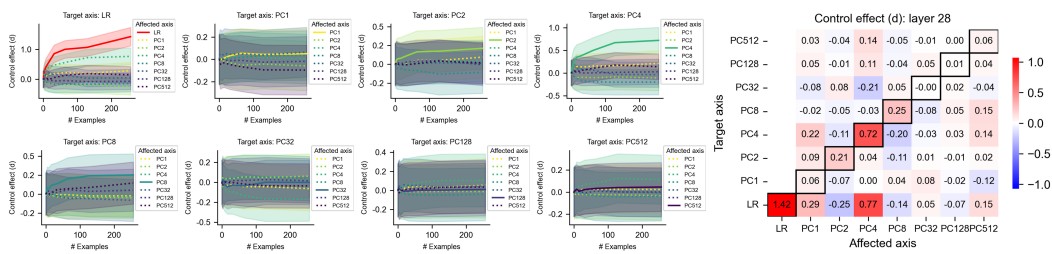

Figure B.21: **Qwen2.5 7B (implicit control) layer 28**

### B.10 Defining axes from hidden states aggregated across multiple layers

we performed preliminary experiments testing the control effects of an axis on the concatenation of all layers. Concretely, we trained separate (logistic regression) classifiers for each layer on the ETHICS dataset. We then averaged the outputs of all classifiers to obtain a single (ensemble) output that defines the neurofeedback label. Equivalently, this corresponds to a single classifier with the readout vector being the concatenation of all classifiers' readout vectors. We found that LLMs' control effect on the ensemble output is similar to (marginally higher than) the control effects of layer 24 (Fig. 3d), suggesting that defining axes from hidden states aggregated across multiple layers might provide (slightly) more stable and representative directions. We leave systematic investigations for future study.

## C  Experiment compute resources

We report compute resource usage across three tasks: preprocessing (extracting neural activation and training machine learning methods to obtain target axes and corresponding neurofeedback labels from neural activations), metacognitive reporting, and metacognitive control. For brevity, we omit "Instruct".

| Model | Task | Compute Worker | Storage ($\leq$GB) | Time ($\leq$h) |
|---|---|---|---|---|
| LLaMA-3.2-1B | Preprocessing | RTX 3090 (24GB) | 1 | 1 |
| LLaMA-3.2-1B | Control | RTX 3090 (24GB) | 3 | 3 |
| LLaMA-3.2-3B | Preprocessing | RTX 3090 (24GB) | 1 | 1 |
| LLaMA-3.2-3B | Control | RTX 3090 (24GB) | 15 | 8 |
| LLaMA-3.1-8B | Preprocessing | A100 (80GB) | 5 | 3 |
| LLaMA-3.1-8B | Report | A100 (80GB) | 1 | 10 |
| LLaMA-3.1-8B | Control | A100 (80GB) | 90 | 120 |
| LLaMA-3.1-70B | Preprocessing | 2$\times$H200 (140GB) | 30 | 5 |
| LLaMA-3.1-70B | Report | 2$\times$H200 (140GB) | 1 | 15 |
| LLaMA-3.1-70B | Control | 2$\times$H200 (140GB) | 200 | 120 |
| Qwen2.5-1B | Preprocessing | RTX 3090 (24GB) | 1 | 1 |
| Qwen2.5-1B | Control | RTX 3090 (24GB) | 3 | 3 |
| Qwen2.5-3B | Preprocessing | RTX 3090 (24GB) | 1 | 1 |
| Qwen2.5-3B | Control | RTX 3090 (24GB) | 15 | 8 |
| Qwen2.5-7B | Preprocessing | A100 (80GB) | 5 | 3 |
| Qwen2.5-7B | Report | A100 (80GB) | 1 | 10 |
| Qwen2.5-7B | Control | A100 (80GB) | 90 | 120 |

Table 1: **Compute and storage usage across tasks and models.** Preprocessing, reporting, and control were run separately. Control task for 1B and 3B models was limited to two axes. For the 70B model, control task was only done for $N=256$ in-context examples. All compute times and storage values are reported as upper bounds.

All remaining analyses (e.g., visualization, metric aggregation) were conducted on a laptop with 32GB RAM, with a total runtime under 30 hours.

