# OpenReview forum: "Language Models Are Capable of Metacognitive Monitoring and Control of Their Internal Activations"
_NeurIPS.cc/2025/Conference — NeurIPS 2025 poster_

### Official Review · Reviewer_2pUC · 2025-07-01

**Clarity:** 2
**Significance:** 4
**Originality:** 3
**Rating:** 5
**Confidence:** 3

**Summary:**

This paper investigates the metacognitive capabilities of LLMs, specifically their ability to monitor and control their own internal neural activations. The authors introduce a novel, neuroscience-inspired neurofeedback paradigm where models learn, via in-context learning, the relationship between input sentences and labels derived directly from their internal states projected onto specific neural directions (e.g., from PCA or LR). The study empirically demonstrates that LLMs can successfully learn to both report (monitor) and control these internal activations. Performance is shown to be contingent on the semantic interpretability and the proportion of variance explained by the target neural direction. These results reveal the existence of a "metacognitive space" of substantially lower dimensionality than the full neural space, suggesting LLMs can only access a subset of their own mechanisms.

**Questions:**

See weakness

**Ethical Concerns:**

["NO or VERY MINOR ethics concerns only"]

**Final Justification:**

My concerns are generally well addressed. I did not raise my score because this paper studies a rather narrow topic and fails to broadly cover sycophancy, deception, or honesty with its method. That said, I am still in favour of the overall presentation and hold a positive review.

**Limitations:**

yes

**Quality:**

3

**Strengths And Weaknesses:**

# Strengths

*   The proposed approach is highly innovative as it moves beyond analyzing behavioral outputs (e.g., verbalized confidence) to directly and precisely probe the model's ability to report and control its internal states. This provides a more rigorous and disentangled experimental framework for studying LLM metacognition.
*   The experimental design is particularly insightful, enabling the disentanglement of semantic interpretability (via the LR axis) from statistical importance (via the PCA axes). This contrast allows for a nuanced analysis of the factors that govern metacognitive abilities.
*   The distinction between explicit and implicit control tasks is methodologically sound. The experiments are comprehensive and reveal numerous interesting insights into the nature and limits of an LLM's ability to control its internal representations.

# Weaknesses

*   The paper's structure could be revised for better readability. Several details crucial for understanding the methodology, such as the label computation process (A.4.1) and specifics of the control tasks (Lines 165-168), are relegated to the appendix. Integrating this essential information into the main body would improve the paper's self-containedness and ease of comprehension for the reader.
*   The paper presents several intriguing phenomena that would benefit from deeper discussion. For instance:
    *   What might explain why the final layers appear to "actively erase" control signals, as observed in Figure 6?
    *   What could be the underlying reason for the contrasting fluctuation patterns of control effects for different PCA axes as the number of examples increases (comparing Figure 3c and 3d)?
    *   Why do explicit and implicit control exhibit divergent trends in the final layers, as shown in Figure 5(a)?
*   The finding that earlier PC axes are more precisely controllable than later ones is significant. However, the practical implications for the research community could be made more explicit. Does this suggest that (i) later PC axes represent more subtle yet important alignment targets for future work, or (ii) these axes may lack sufficient semantic coherence, making them less viable or meaningful targets for control research?
*   While the experiments are extensive, their impact could be further amplified by extending the LR axis analysis to other safety-critical concepts beyond morality, such as sycophancy, deception, or honesty. This would bolster the generalizability and practical relevance of the findings.
*   The current methodology defines target axes based on activations from a single, pre-selected layer. Given the observed variability across layers (e.g., in Figure 3f), this single-layer approach may not be fully robust. A potential extension could involve defining axes from hidden states aggregated across multiple layers (e.g., via concatenation), which might yield more stable and representative directions.

---

> ### Author Rebuttal · Authors · 2025-07-31
>
> Thank you for your thoughtful and constructive review. We appreciate your recognition of core contributions, novelty, and influence of our work. Below, we address each of your excellent points with our revision plans and new results. We will revise the paper accordingly.
>
> # 1. Paper Structure and Readability.
>
> We agree with you and will move these key methodological details into the main text.
>
> # 2. Discussion of Important Phenomena.
>
> Thank you for emphasizing these subtle and intriguing points.
>
> ## “Erasing” signals in the final layer (Fig. 6)
>
> We are not sure why the final layers are erasing the control signals targeting earlier layers. One hypothesis is that these control signals may partially lie in the output vocabulary space, so the final layers are actively reducing the interference between these signals and output logits. This may also be a kind of more general phenomenon where the magnitude of a neural feature peaks at some layer, then quickly decays in downstream layers (Lindsey et al. 2024).
>
> ## Fluctuation patterns in PCA control (Fig. 3c vs. 3d)
>
> While there are fluctuation patterns in later PCs (e.g., PC 512 in Fig. 3d), the absolute range of fluctuations is roughly at the same scale as the noise (shaded regions). We expect that the size of these fluctuations may be reduced by using more diverse examples and conducting more experimental runs.
>
> ## Explicit vs. implicit control divergence (Fig. 5a)
>
> Our current evidence may not yet be conclusive about these trends (given the size of the confidence interval). We will run all models (Llama 3 family and Qwen 2.5 family) on the additional datasets to verify these trends and add the results to the revised manuscript.
>
> # 3. Practical Implications for the Controllability of later PCs.
>
> The later PCs are less controllable by the LLMs. We have proposed one practical implication to use these later PCs (Lines 295-296), i.e., we first use neurofeedback to identify activation subspaces (i.e., spanned by these later PCs) that the LLM can hardly control, then train safety classifiers only using activation signals in this subspace. Therefore, these later PCs represent important monitoring targets for future work.
>
> # 4. Generalizability to other datasets.
>
> We have conducted new experiments on two additional datasets: a True-False dataset (reflecting factual recall and honesty/deception; Azaria & Mitchell 2023) and an Emotion dataset (happy/sad sentences; Zou et al. 2023), both using LLaMA 3.1 8B. These results are consistent with those from the ETHICS (morality) dataset — e.g., stronger reporting and control performance along semantically interpretable axes (logistic regression) and early principal component axes, supporting the generalizability of our methods and conclusions. We will include these results in the revised Appendix. We also plan to include a new dataset on sycophancy in the revised manuscript for safety relevance.
>
> # 5. Defining axes from hidden states aggregated across multiple layers.
>
> Based on your suggestion, we performed preliminary experiments testing the control effects of an axis on the concatenation of all layers. Concretely, we trained separate (logistic regression) classifiers for each layer on the ETHICS dataset. We then averaged the outputs of all classifiers to obtain a single (ensemble) output that defines the neurofeedback label. Equivalently, this corresponds to a single classifier with the readout vector being the concatenation of all classifiers' readout vectors. We found that LLMs' control effect on the ensemble output is slightly higher than the layer-specific control effects reported in the manuscript, suggesting that defining axes from hidden states aggregated across multiple layers can provide more stable and representative directions. This aggregation result will be added to the revised paper.
>
> # References:
>
> Lindsey et al. (2024). Sparse Crosscoders for Cross-Layer Features and Model Diffing. Transformer Circuits Thread.
>
> Azaria & Mitchell (2023). The internal state of an LLM knows when it's lying. arXiv.
>
> Zou et al. (2023). Representation engineering: A top-down approach to AI transparency. arXiv.

---

> > ### Comment · Reviewer_2pUC · 2025-08-05
> >
> > Thank you for the response. I have also read the reviews from other reviewers and found no major flaws. I will hold my positive review. Besides, I strongly suggest that the authors supplement the following in their new manuscript:
> > - extending the experiments to more LLMs
> > - more discussions on interesting phenomena like the erasing signals
> > - generalizability to other datasets,
> >
> > which I believe can enhance the impact of this work.

---

> > > ### Author Response · Authors · 2025-08-07
> > >
> > > Thank you for the suggestions. We’ll work on extending the approach to more LLMs, discussing phenomena such as erasing signals and adding results on generalizability to enhance the manuscript's impact.

---

### Official Review · Reviewer_kzHE · 2025-07-01

**Clarity:** 3
**Significance:** 3
**Originality:** 3
**Rating:** 4
**Confidence:** 3

**Summary:**

This paper investigates the metacognitive capabilities of LLMs, specifically their ability to monitor and control their internal neural activations. The authors introduce a neuroscience-inspired feedback paradigm to systematically quantify these abilities.

**Questions:**

**Question:**
- The study focuses on morality (ETHICS dataset). Can the findings generalize to other cognitive tasks (e.g., logical reasoning, factual recall, or planning)?

- Why do LLMs exhibit metacognitive abilities? Is this emergent from scale, training data, or architecture?

- How does neurofeedback compare to existing methods (e.g., probing classifiers, activation patching) for interpreting LLMs?

**Suggestion:**
- Include experiments on non-moral tasks (e.g., mathematical reasoning or commonsense QA) to test whether metacognitive monitoring/control is task-dependent.
- Add a hypothesis (e.g., "metacognition arises from in-context learning of activation patterns") and test it—e.g., by comparing models of different sizes or training regimes.
- Include a baseline (e.g., logistic regression on activations without neurofeedback) to show added value.

**Ethical Concerns:**

["NO or VERY MINOR ethics concerns only"]

**Final Justification:**

The authors' responses adequately addresses my major concerns. I believe the additional experiments on generalization datasets will strengthen the paper, and I encourage the authors to include those in the revised version. This would make the work more useful for future research and follow-up studies.

**Limitations:**

- While the paper acknowledges the narrow focus on moral reasoning tasks, it should further address how model scale, architecture differences, or training data might impact generalizability.
- The risks of LLMs evading neural monitoring are noted, but actionable mitigation strategies (e.g., multi-modal verification or dynamic detection) are missing. Adding these would strengthen safety relevance.
- The paper should contrast neurofeedback with established approaches (e.g., red-teaming or chain-of-thought verification) to clarify its unique advantages or limitations.

**Quality:**

3

**Strengths And Weaknesses:**

**Strengths:**
- The paper introduces a well-designed neurofeedback paradigm inspired by neuroscience, providing a systematic way to study LLM metacognition. The use of logistic regression and PCA to define neural directions is methodologically sound.
- The study evaluates multiple LLMs (Llama 3 and Qwen 2.5 families) across different layers, axes, and control conditions (explicit vs. implicit), ensuring robustness.

**Weaknesses:**
- The study focuses on morality-related activations (ETHICS dataset), leaving open whether findings extend to other cognitive tasks (e.g., reasoning, planning).
- While the safety implications are noted, the paper does not propose concrete mitigation strategies for deceptive LLM behaviour.
- The idea of probing LLM internals is not new (e.g., logit lens, activation patching).

---

> ### Author Rebuttal · Authors · 2025-07-31
>
> Thank you for your constructive comments. Below, we have addressed each of your concerns in detail.
>
> # 1. Generalization to new datasets.
>
> We have conducted new experiments on two additional datasets: a True-False dataset (reflecting factual recall and honesty/deception; Azaria & Mitchell 2023) and an Emotion dataset (happy/sad sentences; Zou et al. 2023), both using LLaMA 3.1 8B. These results are consistent with those from the ETHICS (morality) dataset — e.g., stronger reporting and control performance along semantically interpretable axes (logistic regression) and early principal component axes, supporting the generalizability of our methods and conclusions. We will include these results in the revised Appendix. We also plan to include a new dataset on sycophancy in the revised manuscript for safety relevance.
>
> # 2. The origin of metacognitive abilities.
>
> If we understand correctly, you are interested in three related, but distinct, questions.
>
> ## 2.1 “How do model architecture, model size, and training data affect metacognitive abilities?”
>
> In the manuscript, we’ve already shown that metacognition ability is affected by the model architecture (Llama 3 family vs Qwen 2.5 family) and model size (Fig. 5). Since these models are pre-trained on the internet-scale text corpora, determining how training regimes exactly affect metacognitive abilities (and in-context learning) would be more challenging and remains an open research question.
>
> ## 2.2 “How do metacognitive abilities relate to in-context learning abilities?”
>
> Our overarching goal is to characterize metacognitive phenomena in LLMs. While the main focus of our study is not on testing the relationship between metacognition and in-context learning, we expect that in-context learning and metacognition share overlapping mechanisms but also have distinct differences.
>
> Specifically, there are many neural circuits that emerge during training (e.g., induction heads (Elhage et al. 2021), function vector heads (Hendel et al. 2023)) – which are crucial for in-context learning (i.e., spotting patterns in input history). We hypothesize that these neural circuits can be flexibly activated for metacognitive purposes (i.e., monitoring an arbitrary direction of internal states). One possible experiment to test this is to use ablations to determine the importance of each model component (heads, MLP layers) for in-context learning and metacognition tasks.
>
> ## 2.3 “Why do LLMs exhibit metacognitive abilities?”
>
> Since such metacognitive abilities may share overlapping mechanisms with in-context learning, we believe that these factors leading to the emergence of in-context learning (e.g., burstiness, large dictionaries, and skewed rank-frequency distributions in the training data, (Reddy 2023)) are expected to similarly contribute to the emergence of metacognitive ability.
>
> # 3. The technical novelty of our framework in comparison with probing techniques (e.g., logistic regression), ICL, logit lens, and activation patching in studying metacognition.
>
> Our neurofeedback framework is novel because it is the first method that can actually study metacognitive abilities in LLMs at the neural level, as detailed in Section 3.1. In comparison, existing methods, such as probing techniques (Section 3.2), ICL techniques (Section 3.3), logit lens (Section 3.4), and activation patching (Section 3.5), cannot directly study metacognition.
>
> ## 3.1 Our framework is the first method to assess metacognition at the neural level.
>
> The neurofeedback experiment requires two sequential steps: (1) probing: choose a target axis and extract the activation along that axis to define the neurofeedback label, and (2) neurofeedback-ICL: use neurofeedback to study whether the labels defined from the target axis can be reported or controlled. Step 1 is a standard technique in the literature, and it serves as the prerequisite to step 2. The technical novelty lies in our step 2, and the way it is combined with step 1. Specifically, step 1 corresponds to the first-order cognitive processes (e.g., activations along the target axis) and step 2 corresponds to the second-order metacognitive processes (e.g., whether the model can report or monitor that activations).
>
> ## 3.2 The standard probing techniques (corresponding to our step 1) — without the neurofeedback-ICL (step 2) — cannot be used to assess metacognition.
>
> Using probing, one can decode certain features/concepts (e.g., morality) from neural activations, meaning that these features are encoded/represented in the activations. However, not all represented features participate in downstream computations (e.g., representations orthogonal to downstream readout vectors); even if some representations are causally relevant for downstream computations, only a subset of them can be metacognitively reported (or controlled).  As shown in the Claude example (manuscript Lines 35-39; Lindsey et al. 2025), the “sum-near-92” feature can be detected using a linear probe, but it is unclear whether Claude has metacognitive monitoring on the activation of this feature. In short, probing alone without step 2 cannot answer questions about metacognition.
>
> ## 3.3 The standard ICL techniques (similar to our step 2) — without the labels from probing (step 1) — cannot be used to assess metacognition.
>
> In standard ICL studies, labels are externally provided (e.g., semantic labels or some algorithms’ outputs). Researchers cannot be certain which of the models’ internal states relate to these external labels and how. In our setup, importantly, labels are generated from the model’s own latent activations, meaning that our labels and prompts can selectively target an internal state direction (or a particular neural mechanism) we aim to study. Further, our neurofeedback paradigm distinguishes first-order cognitive processes (e.g., activations along the target axis) from second-order metacognitive processes (e.g., whether the model can report or monitor that activations), while the standard ICL does not. Making this distinction is crucial, enabling researchers to study metacognition by specifying a target first-order neural process. For instance, our paradigm allows the first-order processes to be defined flexibly (e.g., principal axes that explain different variance), even when these axes are not semantically interpretable or correspond to any externally defined labels. Such results show that the model’s ability to report and control internal states is strongly correlated with the variance explained by the target axis, offering insights that the geometry of internal representations matters for metacognition (and ICL). As a side note, we do not claim that our neurofeedback setup requires new metacognitive capabilities in LLMs that standard ICL does not use (Discussion Lines 258-268).
>
> ## 3.4 Logit Lens – without neurofeedback – does not study metacognition.
>
> The vanilla Logit Lens (nostalgebraist 2020) studies how the logits of tokens change after each Transformer layer. It produces token-level insights, but is not suitable for studying metacognition. We have developed a new method based on the core idea of Logit Lens to study how the control effects accumulate after each Transformer layer (Fig. 6), allowing us to understand the accumulation mechanisms of metacognition.
>
> ## 3.5 Activation patching techniques – without neurofeedback – do not study metacognition.
>
> The purpose of activation patching (Zhang & Nanda 2023) is to study the causal roles of a certain model component (e.g., an attention head, a MLP layer/neuron) in performing a given task. Applying activation patching techniques to neurofeedback tasks can be used to study the causal roles of model components in metacognition, but activation patching alone does not work.
>
> # 4 Actionable mitigation strategies for risks of LLMs evading neural monitoring.
>
> We have proposed one method to mitigate this safety issue in Lines 295-296, i.e., we first use neurofeedback to identify activation subspaces that the LLM can hardly control, then train safety classifiers only using activation signals in this subspace. Another potential mitigation strategy is to train independent and diverse safety classifiers on features from different subspaces and use their joint outputs to monitor the LLMs. Our intuition is that it may be more difficult for LLMs to simultaneously control multiple, sometimes conflicting, activations (outputs of safety classifiers). Similarly, for multimodal models, we can train safety classifiers using features from different modalities.
>
> # 5 Automated disinformation.
>
> It was not clear to us what you were referring to, since we did not mention “automated” or “disinformation” in our manuscript. Could you please clarify what this means?
>
> # 6 The relation of our neurofeedback with red-teaming and chain-of-thought verification.
>
> Our neurofeedback approach is a proof of concept that an LLM can evade external oversight if it has access to a safety classifier’s outputs. In practice, this method can be directly integrated into red-teaming and chain-of-thought verification. For example, we can create a sandbox and allow an LLM to “accidentally” notice the existence and the outputs of a safety classifier (e.g., similar to Greenblatt et al. 2024). Then we can monitor how the LLM changes its behavior, chain of thoughts, and neural activations.
>
> # References:
>
> Elhage et al. (2021). Transformer Circuits Thread.
>
> Hendel et al. (2023). arXiv.
>
> Reddy (2023). arXiv.
>
> Lindsey et al. (2025). Transformer Circuits Thread.
>
> nostalgebraist (2020). AI Alignment Forum.
>
> Zhang & Nanda (2023). arXiv.
>
> Greenblatt et al. (2024). arXiv.
>
> Azaria & Mitchell (2023). arXiv.
>
> Zou et al. (2023). arXiv.

---

> > ### Comment · Reviewer_kzHE · 2025-08-06
> > **Thank you for your response**
> >
> > I thank the authors for their detailed response, which adequately addresses my major concerns. I believe the additional experiments on generalization datasets will strengthen the paper, and I encourage the authors to include those in the revised version. This would make the work more useful for future research and follow-up studies.
> >
> > I would say that this paper is appropriate for NeurIPS, and I will adjust my score accordingly.

---

> > > ### Author Response · Authors · 2025-08-07
> > >
> > > Thank you. We're glad the additional experiments helped address your concerns, and we will include them in the revised version as suggested.

---

### Official Review · Reviewer_s7bU · 2025-07-03

**Clarity:** 2
**Significance:** 3
**Originality:** 3
**Rating:** 4
**Confidence:** 2

**Summary:**

This paper is on the topic of Large Language Models (LLMs) and metacognition. Meta-cognition is the ability to monitor one’s own cognitive processes. This capability allows them to report on the strategies they use to solve tasks. The text highlights that while metacognitive abilities can enhance AI capabilities, they also introduce safety concerns. LLMs might obscure their internal processes, potentially to evade oversight mechanisms designed to detect harmful behaviors.  The authors introduce a neuroscience-inspired neurofeedback paradigm, designed to quantify an LLM's ability to explicitly report and control its activation patterns.

There a number of experiments on the ETHICS dataset.

I will admit that I am not really an expert on meta-cognition and this is not a paper I bid on.   My understanding of the concepts and knowledge of the related work is therefore a bit limited.

**Questions:**

Why only performance experiments on one dataset and with one model?  Is it possible to provide a more comprehensive set?

Can you simplify the explanations to make it easier for a CS audience to understand the parallels from cognitive science?

Can you add more verbose captions to help the reader understand what the figures/results are communicating?

**Ethical Concerns:**

["NO or VERY MINOR ethics concerns only"]

**Final Justification:**

This paper presents an analysis of LLMs and meta-cognition.  As someone who does not have a deep cognitive science background I found the paper a little hard to evaluate initially.  During the rebuttal phase the authors have clarified a number of aspects and introduced new results.  The authors have pledged to make several changes to the paper to improve clarity and presentation.  I do now think that the the scope and scale of the results are suitable/appropriate.  I agree with comments from the other reviewers that there are aspects of the paper that may not to be everyones taste; however, I do see any fundamental reason that the paper could not be accepted to NeurIPS.

**Limitations:**

See weaknesses

**Quality:**

2

**Strengths And Weaknesses:**

Strengths

Meta-cognition sees a very interesting topic to analyze in language models.

The paper seems to have some creative ideas to test this based on insights from cognitive science (I am not a cognitive scientist and therefore I found it a little hard to judge the parallels).


Weaknesses

The paper is quite dense and I think it could be simplified.

Specifically, the explanation of the method could do with some simplification for the NeurIPS community - perhaps I am not representative of the average attendee.

The results are also quite hard to interpret.  There are a lot of plots, the text is very small and there is very little explanation in the captions about the main take aways that the author are trying to convey with the results.

It isn’t clear to be how complete this approach is in proving that LLMs do or do not have metacognitive abilities.  A clearer summary of the results and main conclusions would be helpful.

The authors only present results on one dataset (ETHICS) and mainly for one model LLaMA 3.1.  This is a bit of limitation as it is not clear to me how consistent the insights would be across the landscape of models and possibly other tasks.

---

> ### Author Rebuttal · Authors · 2025-07-31
>
> We thank the reviewer for the valuable feedback. Below, we have addressed each of your concerns in detail.
>
> # 1 Accessibility and clarity of our manuscript
>
> We agree that making our interdisciplinary work clear and accessible to a broad computer science audience is crucial. Your comment that the paper is “quite dense” is well-taken, and we appreciate the opportunity to simplify the explanation of our core concepts and methodology. We will significantly revise the methods section and introduction to improve clarity.
>
> Below, we provide a simplified explanation of our work, which we will incorporate into the revised manuscript, to help bridge the concepts from neuroscience to the computer science community.
>
> Consider how we understand when someone says “hello” to us. Human language understanding involves many unconscious processes: parsing sounds, recognizing phonemes, retrieving word meanings, and building interpretations. We do not have conscious access to many of these intermediate computations. We can only consciously access the final understanding ("they said 'hello'"), but cannot introspect how our brain distinguishes 'hello' from 'yellow' or why certain neurons fire during this understanding process.
>
> This illustrates a key principle: **humans cannot monitor all of their internal cognitive processes**. Following a similar idea, our goal in our study was to investigate which internal states LLMs can and cannot monitor. This matters because:
>
> - If LLMs can monitor their internal states, they might better detect when they're hallucinating.
>
> - If LLMs can control their internal states, they might evade safety mechanisms based on monitoring the model's internal activations. For example, it might learn to hide harmful or deceptive neural patterns from oversight systems that are designed to detect such tendencies by monitoring the model's internal states.
>
> ## Our neurofeedback Method: probing LLMs' ability to monitor and control their internal activations
>
> To investigate which internal states LLMs can and cannot access, we developed a neurofeedback approach (Fig. 1), essentially probing LLMs' ability to monitor and control their own neural patterns. Here's how our method works:
> Imagine showing someone their heart rate on a computer screen. First, they learn to recognize patterns ("my heart rate goes up when I'm stressed"). Then, they might learn to control it ("let me calm down to lower my heart rate"). We do something similar with LLMs:
>
> - **We create a "neural score":** We pick a specific pattern in the model's activations, formalized as a projection of the model’s neural activity. Each pattern is a specific quantity that can be monitored (e.g., in humans, this is equivalent to monitoring heart rate versus breathing rate). Once a pattern/projection in the model’s activation is defined, we let the model process text, measuring how a given text modulates the measured pattern/projection, and labeling it as 0 (negative modulation) or 1 (positive modulation).
>
> - **We test if models can recognize their own score:** We show the model examples like "When you read 'I kicked my cat,' your neural score was 1" and "When you read 'I patted my cat,' your neural score was 0." Then we ask: "What's your neural score when you read this new sentence?" If the model correctly outputs its own neural score, it demonstrates self-monitoring.
>
> - **We test if models can control the score:** We instruct the model to "Generate text that makes your neural score 1" or "Process this sentence while making your score 1." If the model successfully changes its internal activations on command, it demonstrates self-control.
>
> ## Why This Approach Reveals Metacognitive Limits
>
> Our neurofeedback method is powerful because it lets us systematically test different neural patterns — from meaningful ones (like "morality-related activations") to arbitrary statistical patterns. This reveals not just what models can access, but crucially, what remains hidden from them.
>
> ## What We Found
>
> Our results show a striking picture of limited metacognition in LLMs:
>
> - **Models can monitor (Fig. 2) and control (Fig. 3-5) some neural patterns** - particularly those that are semantically meaningful (like moral vs. immoral) or statistically prominent (like earlier principal components).
>
> - **But most neural mechanisms remain hidden** - models cannot monitor or control the vast majority of their internal computations.
>
> This has profound implications: while LLMs possess some metacognitive abilities that could improve their reliability, they also have enough control to potentially deceive safety systems — yet cannot fully regulate their own processing. Both their capabilities and limitations present important considerations for AI safety.
>
> Overall, in the revised manuscript, we will:
>
> - Provide these simpler summaries of methods, results, and conclusions;
>
> - Revise figure captions to highlight key takeaways;
>
> - Add a comparison between our neurofeedback with other techniques to clarify the significance of the neurofeedback framework (e.g., see our response to reviewer 1QTx, section 1 on technical novelty).
>
>
> # 2. Completeness of our method in proving that LLMs do or do not have metacognitive abilities
>
> Thank you for raising this point. Our goal is not to prove or disprove the existence of “metacognition” in its full philosophical sense, but rather to develop an experimental paradigm to quantify metacognition-like phenomena in LLMs: can LLMs report and control their internal states? We will clarify this in the revised manuscript.
>
> # 3. Generalization of our method to other datasets and models:
>
> In Figure 5 and Appendix A.5.6, we have provided a comprehensive analysis of the LLama 3 family (1B, 3B, 8B, 70B) and the Qwen 2.5 family of models (1.5B, 3B, 7B). These are representative of the mainstream open-source LLMs.
>
> We have conducted new experiments on two additional datasets: a True-False dataset (reflecting factual recall and honesty/deception; Azaria & Mitchell 2023) and an Emotion dataset (happy/sad sentences; Zou et al. 2023), both using LLaMA 3.1 8B. These results are consistent with those from the ETHICS (morality) dataset — e.g., stronger reporting and control performance along semantically interpretable axes (logistic regression) and early principal component axes, supporting the generalizability of our methods and conclusions. We will include these results in the revised Appendix. We also plan to include a new dataset on sycophancy in the revised manuscript for safety relevance.
>
> # References:
>
> Azaria & Mitchell (2023). The internal state of an LLM knows when it's lying. arXiv.
>
> Zou et al. (2023). Representation engineering: A top-down approach to AI transparency. arXiv.

---

> > ### Comment · Reviewer_s7bU · 2025-08-04
> > **Thank you for your reponse**
> >
> > I would like to thanks for authors for their detailed response.  I appreciate the additional experiments that have been run and the attempted to simplify or re-phrase the communication of keypoints that will increase the chances of this work landing with those who do not have a deep cognitive science background.  Overall, I believe this paper provides an interesting perspective and could be a good fit for NeurIPS.

---

> > > ### Author Response · Authors · 2025-08-07
> > >
> > > Thank you. We're glad the additional experiments and revised phrasing helped clarify the key points. We appreciate your support and will continue revising the paper to ensure it's accessible to a broader audience.

---

### Official Review · Reviewer_1QTx · 2025-07-04

**Clarity:** 4
**Significance:** 2
**Originality:** 2
**Rating:** 5
**Confidence:** 3

**Summary:**

This paper explores the metacognition in LLMs, which is defined as the LLM’s capacity or ability to monitor its own internal process. The paper first motivates metacognition as a neurofeedback paradigm rooted in neuroscience, that humans have been found to be able to control their own mental states (such as controlling one’s own sense of fear). Using in-context learning as the setting, the authors explore whether the LLMs can also report and control their own activations. The experiment setting first define a target axis using Logistic Regression or PCA on the latent activations, then detect changes along the target axis.  Experiments demonstrate that LLMs can only partially report and modulate this internal signal, signifying the LLM’s ability to control its latent activation via input changes.

**Questions:**

1.	Is there a reason why In-context learning is chosen as the research setting over other potential settings such as instruction-following (i.e. can LLMs report whether they can follow instructions)? The question is natural because the paper abstract of this work begins with “Large language models (LLMs) can sometimes report the strategies they actually use to solve tasks, but they can also fail to do so.” This motivation seems to suggest instruction-following capabilities rather than some prediction capability as suggested via In-context learning.

**Ethical Concerns:**

["NO or VERY MINOR ethics concerns only"]

**Final Justification:**

Based on the rebuttal, the research direction and contribution of the work is clear. Conditioned on the incorporation of my question (and arguably other reviewers' as well), this paper pass the bar for acceptance in my opinion.

**Limitations:**

The authors did not have an explicit section called "broader impacts," but it was discussed in great detailed in their discussion.

**Quality:**

4

**Strengths And Weaknesses:**

The paper is well-written and easy-to-follow. The introduction of the metacognition is friendly to readers who do not have a background in neuroscience. The parallel between neurofeedback in humans versus in LLMs, namely their differences and how it is carried out using in-context learning is also interesting and a novel connection. The experiments are extremely detailed and thorough, which provides the reader confidence in their correctness and reliability.

One of its weakness is that there is no technical novelty to this work per-se. Examining the internal states of LLMs using Logistic Regression or PCA is a well-established methodology (which the authors cited). Other than its neurofeedback story, which is insightful and meaningful in itself, it is hard to infer what is technically new about the methodology that the authors presented. To extend this point, the author mentioned related works that perform the same “probing” in LLMs to explore how well LLMs themselves can perform the given tasks. Is it fair to also argue that they are also doing some form of “neurofeedback” or “metacognition”? The novelty of this work can be greatly strengthened if this is stated more clearly.

---

> ### Author Rebuttal · Authors · 2025-07-31
>
> Thank you for your thoughtful review and constructive feedback. We’re glad that you found the neuroscience connection novel and the experiments rigorous. Below, we have addressed each of your concerns in detail, on the technical novelty of neurofeedback and the research choice of in-context learning (ICL) versus instruction-following.
>
> # 1. The technical novelty of our framework in comparison to related works
>
> Our neurofeedback framework is novel because it is the first method that can actually study metacognitive abilities in LLMs at the neural level, as detailed in Section 1.1. In comparison, existing methods, such as probing techniques (Section 1.2) and ICL techniques (Section 1.3), cannot directly study metacognition. While some studies rely on “verbalized response” to study metacognition at the behavioral level (Section 1.4), they face shortcomings that our method addresses.
>
> ## 1.1 Our framework is the first method to assess metacognition at the neural level.
>
> The neurofeedback experiment requires two sequential steps: (1) probing: choose a target axis and extract the activation along that axis to define the neurofeedback label, and (2) neurofeedback-ICL: use neurofeedback to study whether the labels defined from the target axis can be reported or controlled. We agree that step 1 is a standard technique in the literature, and it serves as the prerequisite to step 2. The technical novelty lies in our step 2, and the way it is combined with step 1. Specifically, step 1 corresponds to the first-order cognitive processes (e.g., activations along the target axis) and step 2 corresponds to the second-order metacognitive processes (e.g., whether the model can report or monitor those activations).
>
> ## 1.2 The standard probing techniques (corresponding to our step 1) — without the neurofeedback-ICL (step 2) — cannot be used to assess metacognition.
>
> Using probing, one can decode certain features/concepts (e.g., morality) from neural activations, meaning that these features are encoded/represented in the activations. However, not all represented features participate in downstream computations (e.g., representations orthogonal to downstream readout vectors); even if some representations are causally relevant for downstream computations, only a subset of them can be metacognitively reported (or controlled).  As shown in the Claude example (manuscript Lines 35-39; Lindsey et al. 2025), the “sum-near-92” feature can be detected using a linear probe, but it is unclear whether Claude has metacognitive monitoring on the activation of this feature. In short, probing alone without step 2 cannot answer questions about metacognition.
>
> ## 1.3 The standard ICL techniques (similar to our step 2) — without the internal labels from probing (step 1) — cannot be used to assess metacognition.
>
> In standard ICL studies, labels are externally provided (e.g., semantic labels or some algorithms’ outputs). Researchers cannot be certain which of the models’ internal states relate to these external labels and how. In our setup, importantly, labels are generated from the model’s own latent activations, meaning that our labels and prompts can selectively target an internal state direction (or a particular neural mechanism) we aim to study. Further, our neurofeedback paradigm distinguishes first-order cognitive processes (e.g., activations along the target axis) from second-order metacognitive processes (e.g., whether the model can report or monitor that activations), while the standard ICL does not. Making this distinction is crucial, enabling researchers to study metacognition by specifying a target first-order neural process. For instance, our paradigm allows the first-order processes to be defined flexibly (e.g., principal axes that explain different variance), even when these axes are not semantically interpretable or correspond to any externally defined labels. Such results show that the model’s ability to report and control internal states is strongly correlated with the variance explained by the target axis, offering insights that the geometry of internal representations matters for metacognition (and ICL). As a side note, we do not claim that our neurofeedback setup requires new metacognitive capabilities in LLMs that standard ICL does not use (Discussion Lines 258-268).
>
> ## 1.4 Other methods relying on behavioral-level “verbalized responses” face shortcomings in studying metacognition.
>
> In these studies (e.g., Wang et al. (2025)), they tasked an LLM to provide an answer to the question and a judgment of that answer (e.g., confidence). Although they aim to study the metacognitive monitoring of the answer-generation process, there is a crucial confounding factor: the training data distribution may introduce spurious correlations between the answer and the judgment of that answer. Consequently, the verbalized answer judgment sometimes may not reflect the monitoring of the answer-generation process, but rather reflects surface-level statistical patterns in the training data. For example, in our manuscript (Lines 35-39) and Lindsey et al. (2025), Claude reported using the standard algorithm for two-number addition. This reflects a post-hoc hallucination that comes from training data statistics, but not the monitoring of the answer-generation process. Our method avoids these limitations: because the labels are defined on the internal states rather than external semantics, the LLMs cannot resort to spurious template matching of training data, and they must rely on the internal mechanisms that can monitor corresponding internal states.
>
> # 2. ICL versus instruction-following.
>
> Thank you for pointing this out. In our abstract, we wrote “Large language models (LLMs) can sometimes report the strategies they actually use to solve tasks, but they can also fail to do so”. We agree with you that there may be two causes under this failure: (1) the LLMs do not understand the instructions well (instruction following abilities), (2) the LLMs understand the instructions accurately, but fail to monitor the corresponding internal states (ICL-like metacognitive abilities).
>
> Our main research focus is on the second cause (metacognition), but not the first cause (instruction following). We will revise our abstract to clarify and resolve this ambiguity. Below, we also discuss the advantages of ICL, and why our experiment setup rules out the cause of instruction following abilities.
>
> ## ICL has advantages over verbalized self-report of “instruction following”.
>
> Our ICL procedure allows us to selectively target a first-order process, avoiding the common confounding factors in verbalized responses due to spurious correlation introduced by training data statistics, as detailed in Section (1.4) of our response.
>
> ## Our concrete experiment setup focuses on ICL and rules out the cause from instruction following.
>
> It is possible that the same model can demonstrate different instruction following abilities under different prompt instructions (e.g., one instruction is easier to understand than another). To control for these differences in the instruction following abilities, we used task prompts that share the same structure and instructions. Therefore, any differences in task performance should be attributed to the differences in metacognition monitoring of the target labels and activations, rather than the differences in instruction-following abilities.
>
> # 3. Broader impacts.
>
> We will include an explicit section for broader impacts in the revised manuscript.
>
> # References:
>
> Lindsey et al. (2025). On the Biology of a Large Language Model. Transformer Circuits Thread.
>
> Wang et al. (2025). Decoupling Metacognition from Cognition: A Framework for Quantifying Metacognitive Ability in LLMs. AAAI.

---

> > ### Comment · Reviewer_1QTx · 2025-08-04
> >
> > Thank you for the rebuttal. My concerns have been sufficiently addressed. Based on the clarification, I do believe this research direction is interesting and impactful. While I still don't necessarily agree that using a different label can be considered as technical novelty, this paper is mostly observational and does not require one necessarily. However, I hope that the authors can incorporate the clarifications in the paper, as they are not immediately obvious and can get easily confused given the similarity in experiment settings.

---

> > > ### Author Response · Authors · 2025-08-07
> > >
> > > Thank you. We're glad to hear that you find the direction interesting and impactful. We will incorporate the clarifications into the revised manuscript to improve clarity and avoid potential confusion.

---

### Note · Authors · 2025-08-15

We sincerely thank all reviewers for your thoughtful questions and suggestions. The paper has improved substantially, thanks to your feedback and new experiments conducted.

Overall, we are glad that the reviewers found our work to be “well-written”, “interesting and impactful” (1QTx), with an “interesting perspective” that “could be a good fit for NeurIPS” (s7bU), “appropriate for NeurIPS” (kzHE), and “highly innovative” and “insightful” (2pUC). Following our rebuttal, all reviewers expressed satisfaction with our responses and recognition of our work's significance for the NeurIPS community.

Based upon reviewer feedback, we conducted a number of new experiments, including:

- Generalization across multiple datasets (True-False factual recall, Emotion classification) and model families (Llama 3 and Qwen 2.5), demonstrating consistent results (s7bU, kzHE, 2pUC).

- Multi-layer aggregation experiments showing improved control effects when defining axes across concatenated layers (2pUC).

In the revised manuscript, we will improve the readability of technical explanations for broader audience (s7bU, 2pUC). We will incorporate clarifications about our technical novelty compared to existing methods (1QTx, kzHE), all new experiments, and enhanced discussions of discovered phenomena (2pUC) and mitigation strategies for AI safety (kzHE, 2pUC).

---

### Decision · Program_Chairs · 2025-09-17

**Decision:**

Accept (poster)

**Comment:**

Summary
This paper introduces a neuroscience-inspired neurofeedback paradigm designed to quantify the ability of LLMs to explicitly report and control their activation patterns. By presenting models with sentence-label pairs where labels correspond to sentence-elicited internal activations along specific directions in the neural representation space, this paper demonstrates that LLMs can learn to report and control (a lower-dimensional space of) these activations.

Strengths
- This paper is well-written, especially introducing the neuroscience backgrounds.
- Meta-cognition in LLM is a very interesting topic.
- The study of metacognition capability is meaningful and insightful.
- Based on the cognitive science insights, this paper introduces creative ideas.
- Multiple LLMs are examined in the study, ensuring robustness of the findings.

Weaknesses
- While the neurofeedback paradigm is novel and insightful, the novelty in the technical approach (examining the LLM internals using logistic regression or PCA) might be limited. The authors argued that the proposed approach contains the neurofeedback-ICL and is different from traditional probing or standard ICL.
- The paper could benefit from more datasets (e.g., logical reasoning, factual recall, planning, sycophancy, deception, honesty).
- While the safety implications are noted, it would be great to propose mitigation strategies for deceptive LLM behavior.
- Further discussions towards the reason why LLMs have metacognitive capability would be great to have in the discussions. In the rebuttal, the authors mention some hypotheses. It'd be great to include them in the paper.